# “Cell Membrane Theory of Senescence” and the Role of Bioactive Lipids in Aging, and Aging Associated Diseases and Their Therapeutic Implications

**DOI:** 10.3390/biom11020241

**Published:** 2021-02-08

**Authors:** Undurti N. Das

**Affiliations:** 1UND Life Sciences, 2221 NW 5th St, Battle Ground, WA 98604, USA; undurti@lipidworld.com; Tel.: +508-904-5376; 2BioScience Research Centre and Department of Medicine, GVP Medical College and Hospital, Visakhapatnam 530048, India; 3International Research Centre, Biotechnologies of the third Millennium, ITMO University, 191002 Saint-Petersburg, Russia

**Keywords:** aging, unsaturated fatty acids, bioactive lipids, inflammation, cell membrane, sirtuins

## Abstract

Lipids are an essential constituent of the cell membrane of which polyunsaturated fatty acids (PUFAs) are the most important component. Activation of phospholipase A2 (PLA2) induces the release of PUFAs from the cell membrane that form precursors to both pro- and ant-inflammatory bioactive lipids that participate in several cellular processes. PUFAs GLA (gamma-linolenic acid), DGLA (dihomo-GLA), AA (arachidonic acid), EPA (eicosapentaenoic acid) and DHA (docosahexaenoic acid) are derived from dietary linoleic acid (LA) and alpha-linolenic acid (ALA) by the action of desaturases whose activity declines with age. Consequently, aged cells are deficient in GLA, DGLA, AA, AA, EPA and DHA and their metabolites. LA, ALA, AA, EPA and DHA can also be obtained direct from diet and their deficiency (fatty acids) may indicate malnutrition and deficiency of several minerals, trace elements and vitamins some of which are also much needed co-factors for the normal activity of desaturases. In many instances (patients) the plasma and tissue levels of GLA, DGLA, AA, EPA and DHA are low (as seen in patients with hypertension, type 2 diabetes mellitus) but they do not have deficiency of other nutrients. Hence, it is reasonable to consider that the deficiency of GLA, DGLA, AA, EPA and DHA noted in these conditions are due to the decreased activity of desaturases and elongases. PUFAs stimulate SIRT1 through protein kinase A-dependent activation of SIRT1-PGC1α complex and thus, increase rates of fatty acid oxidation and prevent lipid dysregulation associated with aging. SIRT1 activation prevents aging. Of all the SIRTs, SIRT6 is critical for intermediary metabolism and genomic stability. SIRT6-deficient mice show shortened lifespan, defects in DNA repair and have a high incidence of cancer due to oncogene activation. SIRT6 overexpression lowers LDL and triglyceride level, improves glucose tolerance, and increases lifespan of mice in addition to its anti-inflammatory effects at the transcriptional level. PUFAs and their anti-inflammatory metabolites influence the activity of SIRT6 and other SIRTs and thus, bring about their actions on metabolism, inflammation, and genome maintenance. GLA, DGLA, AA, EPA and DHA and prostaglandin E2 (PGE2), lipoxin A4 (LXA4) (pro- and anti-inflammatory metabolites of AA respectively) activate/suppress various SIRTs (SIRt1 SIRT2, SIRT3, SIRT4, SIRT5, SIRT6), PPAR-γ, PARP, p53, SREBP1, intracellular cAMP content, PKA activity and peroxisome proliferator-activated receptor γ coactivator 1-α (PGC1-α). This implies that changes in the metabolism of bioactive lipids as a result of altered activities of desaturases, COX-2 and 5-, 12-, 15-LOX (cyclo-oxygenase and lipoxygenases respectively) may have a critical role in determining cell age and development of several aging associated diseases and genomic stability and gene and oncogene activation. Thus, methods designed to maintain homeostasis of bioactive lipids (GLA, DGLA, AA, EPA, DHA, PGE2, LXA4) may arrest aging process and associated metabolic abnormalities.

## 1. Introduction

Aging is inevitable. It is associated with time-dependent decline in physical activity and physiological function, and slow but steady decrease in organ function that eventually leads to death. With increasing age, aging associated diseases occur that include insulin resistance, type 2 diabetes mellitus, hypertension, hyperlipidemia or dyslipidemia, coronary heart disease, Alzheimer’s disease, and cancer. Aging induces perceptible and time-dependent changes in the immune system and gradual increase in plasma inflammatory markers interleukin-6 (IL-6) and tumor necrosis factor-α (TNF-α) with a parallel decrease in anti-inflammatory cytokines IL-4 and IL-10. However, it is not known whether changes in the levels of cytokines are inevitable with aging or are a signal of the impending aging process or just a marker of aging. It is not clear whether changes in cytokines and associated events such as free radical generation (reactive oxygen species, nitric oxide, carbon monoxide, hydrogen sulfide) and antioxidants are as a result of aging. However, what is certain is that these biochemical and immunological changes can be postponed or delayed by regular exercise and diet control (especially calorie restriction and intermittent fasting). Good dietary practices and regular exercise can delay the development of aging associated diseases obesity, insulin resistance, type 2 diabetes mellitus, hypertension, metabolic syndrome, coronary heart disease and cancer. 

Understanding the molecular events responsible for aging process may help to develop strategies to slow aging. Optimal cell response to both internal and external stimuli depends on the cell membrane integrity. This is so since; all stimuli need to be conveyed to the genome through the cell membrane. Similarly, all the responses elicited by the cell genome need to be conveyed to the cell external milieu through the cell membrane. Thus, cell membrane structure and consequently its functions are crucial to receive and send signals to the external environment. Since aging is a universal process (from single cell organisms to humans), it is possible that understanding the molecular events of this process may have implications for all. Some of the outlined biological mechanisms of the aging proposed are given in Figure 1. Thus, methods or strategies developed to act on these processes may lead to certain common management strategies of diseases associated with aging and eventually ensure delaying or postponing or even preventing aging itself. 

## 2. Cell Membrane Theory of Aging

Both lipids and proteins (and their associated carbohydrate molecules) are important constituents of the cell membrane. Proteins are like bricks of the wall and inflexible. In contrast, lipids are flexible, and they influence cell membrane fluidity. The presence of higher amounts of unsaturated fatty acids render the membrane more fluid whereas higher content of saturated fatty acids and cholesterol make the membrane more rigid. Alterations in the cell membrane fluidity influences the expression of receptors and their affinity to their respective molecules. Hence, the constitution of cell membrane and its lipid content is critical to cell function. 

There are several excellent reviews about the composition of properties of cell membrane in the literature. Hence, no detailed description of the cell membrane is given here. However, it is sufficient to say the following:The cell membrane is the physical and chemical barrier which separates the inside the cell from the outside environment.The structure of the cell membrane can be described as liquid bilayer of lipid embedded with proteins called as a “fluid mosaic model”.This bilayer of the cell membrane is formed by the amphipathic molecules (phosphate rich heads on the outside and hydrophobic lipid tails on the inside).The cell membrane is impermeable to water-soluble molecules but not to water, is soft and flexible. The flexibility of the membrane could be attributed to its lipid content. It has the unique property of being able to spontaneously repair pores.About the composition of the cell membrane: lipids form ~50% by weight, proteins another ~50% by weight and carbohydrate portions of glycolipids and glycoproteins form approximately about 10%.The outer membrane is mainly consisting of phosphatidylcholine and sphingomyelin and the inner membrane is composed of phosphatidylethanolamine, phosphatidylserine and phosphatidylinositol and variable amounts of cholesterol.The transmembrane proteins and lipid-anchored proteins are generally confined to one of the membranes. Most of the receptors for various proteins are located on the outer surface though some receptors are inside the membrane.Glycosylated components of glycolipids and glycoproteins form the carbohydrate component of the membrane and they form the cellular glycocalyx.In general, water is present between lipid molecules in a highly organized form and bulk of the water content is present in the pores and channels.Ions such as calcium, sodium, etc., are present in the membrane and are attracted to the membrane by the intrinsic negative charge of the phospholipid heads.Cholesterol is also a major membrane component and is present in a variable amount, depending on the cell and species.

The cell membrane is crucial for all the cellular functions. Hence, it is reasonable to propose that there could occur a close correlation between cell membrane integrity and changes in its composition to features of aging. This proposal implies that the lipid composition of the cell membrane is critical for the aging process. Of all the lipids that are present in the membrane, unsaturated fatty acids and their metabolites are important not only for cellular functions but also the aging process, implying that lipid composition of the membrane changes with aging and vice versa. At the same time, it is likely that the concentrations of various unsaturated fatty acids and their metabolism varies from cell to cell and is probably specific for each type of cell/tissue/organ. This is supported by the data given in Table 1, which shows that each type of cell/tissue has its own unique lipid composition. Similarly, the concentrations of antioxidants (superoxide dismutase, catalase, glutathione peroxidase), nitric oxide and lipid peroxides have also been reported to be different in various cells/tissues [1,2]. Thus, depending on the cell architecture, function, and its role in various cellular processes, the unsaturated fatty acid content and its metabolism varies. In other words, the composition of the cell membrane and its unsaturated fatty acid content and its metabolism determines the cell function and vice versa. This crosstalk between the cell architecture and function and its membrane composition are integrated with each other in such a way that cell membrane architecture determines the cell function and vice versa. Thus, changes in the lipid composition of the membrane determines the cellular function and its aging process. Since the membrane composition of each cell type is different and specific to it, this explains why different cells/tissues/organs show variation in their aging process. In this review, specific emphasis is given to the metabolism of unsaturated fatty acids and their metabolism (called as bioactive lipids: BALs) and this concept can be extended to other lipids present in the membrane such as phospholipids, cholesterol, glycolipids, phosphoglycerides, sphingolipids, etc. It is noteworthy that of all these lipids, perhaps, unsaturated fatty acids and their metabolism is critical to various cellular functions simply because most of the other lipids such as cholesterol, glycolipids, phosphoglycerides, and sphingolipids can influence the metabolism of unsaturated fatty acids either directly or indirectly. 

In addition to the fact that unsaturated fatty acids composition of cells/tissues are specific to each type of cell/tissue, their concentrations change with aging as well. In a study, where age-related changes in phospholipid fatty acid composition in rat liver, kidney and heart were evaluated (in 3-, 12- and 24-month-old rats) it was noted that saturated fatty acids did not change significantly with age but a significant decrease in LA (18:2 n−6) in the liver and heart, DGLA (20:3 n−6) in the kidney in the organs studied. This change with age in the ratio between saturated and unsaturated n−6 and n−3 fatty acids with the balance tilted more towards the former is likely to produce a decrease in cell membrane fluidity (more rigid). It is noteworthy that dietary restriction significantly reverted these changes that lends support to the concept that decreased calorie intake is beneficial and prolongs lifespan [3,4,5]. Some of the pathways that mediate the beneficial effects of calorie restriction include insulin/insulin growth factor-1 (IGF-1), sirtuins, mammalian target of rapamycin (mTOR), 5′ adenosine monophosphate-activated protein kinase, decrease in ROS generation (reactive oxygen species), increase in antioxidant scavenging capacity, reduced damage to DNA and proteins, increase in autophagy and improvement in T cell function [6,7]. It is likely that all these changes reported with calorie restriction and its beneficial action in delaying aging process could be linked to alterations in the metabolism of essential fatty acids and their impact on cell membrane fluidity and other actions as discussed below. 

BALs produce extensive changes in the membrane fluidity that, in turn, can affect several cellular functions, including but not limited to carrier-mediated transport, the properties of membrane-bound enzymes, expression and binding characteristics of various receptors, phagocytosis, endocytosis, depolarization-dependent exocytosis, immunologic and chemotherapeutic cytotoxicity, prostaglandin production, and cell growth [8,9]. Thus, the actions of bioactive lipids are extraordinarily complex, and they vary from one type of cell to another. It is difficult to make any generalizations or to predict how a given system (cell, tissue, or organ) will respond to a lipid. However, it is likely that many of the functional responses are probably secondary to changes in the membrane structure. It is quite but natural that the conformation or quaternary structures of certain transporters, receptors, and enzymes are sensitive to changes in the structure of their lipid microenvironment, leading to alterations in their activity. The formation and nature of various types of prostaglandins (PGs), thromboxanes (TXs), leukotrienes (LTs), lipoxins (LXs), resolvins, protectins and maresins is certainly modulated by the availability of their precursors that reside in the membrane phospholipids which are released from the membrane stores based on the type, nature, and strength of the stimulus which in itself can cause a change in membrane lipid structure. Thus, a close relationship exists between the membrane lipid compositional change and the concurrent functional perturbations. Since many aging associated diseases are pro-inflammatory in nature (mostly are characterized by low-grade systemic inflammation), it is relevant to delve into the metabolism essential fatty acids (EFAs) and note the various pro- and anti-inflammatory products formed from them. 

## 3. Metabolism of Essential Fatty Acids and Factors That Influence Their Metabolism

PUFAs that form an important constituent of the cell membrane include GLA (gamma-linolenic acid, 18:3 n−6), DGLA (dihomo-GLA, 20:3 n−6), AA (arachidonic acid, 20:4 n−6), EPA (eicosapentaenoic acid, 20:5 n−3) and DHA (docosahexaenoic acid, 22:6 n−3) are formed from dietary linoleic acid (LA, 18:2 n−6) and alpha-linolenic acid (ALA, 18:3 n−3) by the action of desaturases and elongases (see Figure 2) [2,3,4,5]. Both LA and ALA are also considered as PUFAs but are essential fatty acids (EFAs) since they cannot be formed in the body and need to be obtained from diet. GLA, DGLA, AA, EPA and DHA are considered as “functional EFAs” as they can perform some of the functions of LA and ALA. EFA deficiency is rare but can be seen in those who are on total parenteral nutrition if they do not receive enough amounts of LA and ALA. EFA deficiency is characterized by desquamating skin lesions, increased susceptibility to infection, poor wound healing, thrombocytopenia, and growth retardation. LA, ALA, AA, EPA and DHA can be obtained direct from diet and so their deficiency may indicate malnutrition. In such a situation, their deficiency (that of LA, ALA, AA, EPA and DHA) may be associated with deficiency of several minerals, trace elements and vitamins some of which are needed for the physiological activity of desaturases. In the present discussion, PUFAs deficiency states are centered around those who are otherwise healthy though it may be argued that the presence of PUFAs deficiency is in itself a type of malnutrition (where there is consumption of excess of calories or high-fat diet accompanied by subclinical deficiency of minerals, trace elements and vitamins that, in itself may account for decreased formation of GLA, DGLA, AA, EPA and DHA from their respective precursors or decreased dietary intake of AA, EPA and DHA. This is especially true of those who are obese and have insulin deficiency).

Several co-factors involved in the metabolism of EFAs include vitamin B1, B6, B12, folic acid, vitamin C, zinc, magnesium, and calcium. Insulin stimulates delta-6-desaturase and delta-5-desaturaese enzymes and thus enhances the formation of GLA, DGLA, AA from LA and EPA and DHA from ALA. Cholesterol and trans fatty acids inhibit the activity of desaturases and hence, a deficiency of GLA, DGLA, AA, EPA and DHA may occur. Hyperglycemia and hyperlipidemia inhibit the activities of desaturases [10,11,12,13]. Alcohol in the initial stages augments the conversion of DGLA to PGE1, a neurostimulator, but at the same time it inhibits the activity of desaturases. Hence, continuous ethanol consumption leads to a decrease in the concentrations of GLA, DGLA, AA, EPA and DHA due to decreased conversion of LA and ALA to their long-chain metabolites [14]. This action of ethanol on desaturases is in addition to its action on stearoyl-CoA desaturase and palmitoyl-CoA desaturase activities that are decreased, while activities of electron transport components such as NADH-cytochrome c and NADH-ferricyanide reductases are unchanged. Chronic ethanol administration can result in an adaptive induction of the activity of NADPH-cytochrome c reductase and the contents of cytochrome b5 and P-450. The NADH/NAD ratio in microsomes of ethanol-fed rats increased over 2-fold, suggesting chronic ethanol ingestion decreases the activities of delta 9-desaturases. The initial hot flushes due to ethanol consumption has been attributed to increased formation of PGE1. Similarly, nicotinamide also stimulates the formation of PGE1 from DGLA. Ethanol increases triacylglycerol accumulation in the liver, cause direct damage to cell membranes and induce liver steatosis. Increase in saturated fatty acid content and higher levels of LA and ALA in the membrane may occur (due to deficiency of desaturases) that may result in changes in the membrane composition and alteration in its fluidity that may be responsible for the biochemical, physiological and neurobehavioral effects of ethanol. Some, if not all, changes induced by ethanol can be reverted to normal by providing adequate amounts of GLA/DGLA/AA/EPA/DHA that bypass the block in the activities of desaturases. Thus, fetal-alcohol syndrome induced by ethanol can be prevented or ameliorated by providing AA [14,15,16]. These results are in support of the fact that EFAs/PUFAs are needed for brain growth and development [17]. 

Glucose, sucrose, and fructose (fructose > sucrose > glucose) decrease the formation of AA, EPA, and DHA by inhibiting the activities of desaturases. As a result, the formation of lipoxins, resolvins, protectins, and maresins is insufficient [18,19,20,21]. In addition, glucose, sucrose, and fructose have pro-inflammatory actions and enhance the levels of tumor necrosis factor-α (TNF-α) [21]. It is noteworthy that TNF-α itself can cause EFA deficiency state [22] that further enhances TNF-α formation due to the absence of the negative regulation exerted by GLA, DGLA, AA, PGE1, lipoxins, resolvins, protectins, and maresins [23,24,25,26,27,28,29,30].

## 4. Actions of GLA/DGLA/AA/EPA/DHA and Their Metabolites

DGLA/AA/EPA/DHA form precursors to both pro- and anti-inflammatory metabolites. DGLA is the precursor of PGE1, a potent anti-inflammatory, platelet anti-aggregatory, vasodilator and cytoprotective molecule [31,32]. All these fatty acids can alter the membrane fluidity and thus, influence several cellular functions. 

Gamma-linolenic acid (GLA, 18:3 n−6), formed by the action of delta-6-desaturase from dietary LA, has genoprotective action against various chemicals and radiation [31,32]. Similar genoprotective and cytoprotective action is shown by DGLA and PGE1 [31,32]. Thus, GLA, DGLA and PGE1 serve as endogenous cytoprotective and genoprotective molecules in addition to their ability to suppress IL-6 and TNF-α production. In addition, GLA, DGLA and PGE1 have considerable anti-diabetic action as well [33,34]. 

AA is the precursor of 2 series prostaglandins (PGs) and thromboxanes (TXs) and 4 series leukotrienes (LTs), all of which have pro-inflammatory actions except for PGI2 and PGJ2, though this depends on the dose and context. AA is also the precursor of lipoxin A4 (LXA4) that has vasodilator, platelet anti-aggregator, cytoprotective and anti-diabetic actions [30,35,36]. In view of its potent beneficial actions, LXA4 and its precursor AA have the potential to move to the clinic in the prevention and management of inflammatory diseases insulin resistance, obesity, type 2 diabetes mellitus, coronary heart disease, Alzheimer’s disease, inflammatory bowel disease, multiple sclerosis, lupus, rheumatoid arthritis, hypertension, bronchial asthma, non-alcoholic fatty liver disease (NAFLD), alcohol-induced diseases including hepatitis, and cancer that are common in elderly. 

EPA and DHA form precursors to anti-inflammatory compounds resolvins (E series from EPA and D series from DHA and protectins and maresins from DHA) (see Figure 2). EPA forms the precursor of 3 series PGs and TXs and 5 series LTs that also have pro-inflammatory action but much less potent compared to 2 series PG2 and 4 series LTs. These and other results imply that the balance between pro- and anti-inflammatory products formed from AA, EPA and DHA determines the outcome of the inflammatory process. 

## 5. Cross Talk between Pro- and Anti-Inflammatory Molecules

PUFAs are mainly present in the phospholipid (PL) fraction of the cell membrane lipid pool and are released from this pool by the action of phospholipase A2 (and under certain circumstances by the action of PLD and PLC). Several stimuli act on the cell membrane to activate PLA2. There are three classes of phospholipases that control the release of PUFAs (mainly GLA, DGLA, AA, EPA and DHA) from the membrane lipid pool. They are (i) calcium independent PLA2 (iPLA2) that is activated during the first 24 h of exposure to a pro-inflammatory stimulus; (ii) secretary (sPLA2) and cytosolic PLA2 (cPLA2) that are activated during the resolution phase of the inflammation [37,38,39,40,41,42]. There are at least 10 isoenzymes of sPLA2, 3 for cPLA2 and 2 for iPLA2. PUFAs released from the membrane PL pool are acted upon by COX1, COX-2 (cyclo-oxygenase 2) and 5-, 12-, and 15-LOX (lipoxygenase) enzymes to convert them to form PGs, TXs, LTs and lipoxins, resolvins, protectins and maresins from the respective precursors as the situation demands. All these bioactive lipids regulate exudate formation, inflammatory cell influx, and IL-6, TNF-α, HMGB1 (high mobility group box-1), IL-1, IL-2, IL-4, IL-10, IL-17, and IFNs and regulate inflammation and its resolution process. IL-6, TNF-α, PGE2 and LTB4 regulate neutrophil influx process, exudate formation and the initial inflammatory events. In contrast a decrease in PGE2 and an increase in LXA4, PGD2 and its product 15deoxyΔ^12–14^PGJ2 suppress the inflammatory events and induce its resolution [39,40,41,42]. Since both pro- and anti-inflammatory PGs, TXs, LTs, lipoxins, resolvins, protectins and maresins are derived from the same precursors, it is argued that there are two waves of release of AA/EPA/DHA one at the onset of inflammation and a second at the time of resolution of inflammation. The first wave of release of DGLA/AA/EPA/DHA is induced by the activation of iPLA2 (essentially type VI iPLA2) at the onset of inflammation (generally up to 24 h of the onset of inflammation) whereas type IIa and V sPLA2 are activated from the beginning of 48 h to 72 h of inflammation to initiate inflammation resolution process. This increase in type IV cPLA2 is accompanied by a parallel increase in COX-2 expression [41]. This close interaction and enzymatic coupling between COX-2 and PLA2 indicates a specific role for different phospholipases and COX-2 in the inflammatory and resolution processes. IL-1β is involved in both pro- and anti-inflammatory processes [23,37,38,39,40,41,42,43,44]. Activated iPLA2 can convert inactive pro-IL-1β to active IL-1β, which, in turn, induces cPLA2 expression that initiates the resolution. TNF-α and MIF suppress the synthesis of LXs, PGD2, and 15deoxyΔ^12–14^PGJ2 from cPLA2-induced release of AA/EPA/DHA, whereas LXs (especially LXA4), inhibit TNF-α and TNF-α-induced production of ILs; enhance TNF-α mRNA decay, TNF-α secretion, and leukocyte trafficking events that ultimately result in resolution of inflammation. Dexamethasone and other corticosteroids inhibit both cPLA2 and sPLA2 expression, while type IV iPLA2 expression is refractory to their suppressive actions [45,46]. During the normal course of an inflammatory process, the local concentrations of endogenous corticosterone are high, whereas at the time of resolution they are low. Such an alteration in the levels of corticosteroids ensures that cPLA2 and sPLA2 can be expressed to augment the production of LXs, PGD2, and 15deoxyΔ^12–14^PGJ2 to initiate and induce inflammation resolution. This close interaction and positive and negative feedback regulation among PLA2s, corticosteroids, cytokines, COX-2, LOX, PGE2, PGD2, LXA4, and PAF (platelet activating factor) in the inflammatory process implies that any imbalance in this complex process may result in either persistence of inflammation or sub-optimal inflammation resolution that may happen in aging associated diseases (see Figure 3). 

## 6. Pro- and Anti-Inflammatory Actions of PGE2 

In general, PGE2 is considered as a pro-inflammatory molecule. However, under some specific conditions, it behaves as an anti-inflammatory molecule at least, in part, by altering macrophage polarization by mesenchymal stem cells (MSCs) [47,48,49,50,51,52,53,54]. This is because PGE2 can bind to its receptors EP2 and EP4 depending on its concentration. Low PGE2 concentrations bind to the high-affinity EP4 receptor and augment IL-23 production, whereas high concentrations bind to EP2 receptor and inhibit IL-23 release [51,52]. This may explain the dual pro- and anti-inflammatory actions of PGE2. In addition, PGE2 induces the production of LXA4 (by stimulating the conversion of AA to LXA4) and simultaneously inhibits LTB4 synthesis by modulating 5- and 15-lipoxygenase expression. This increase in LXA4 production and a reduction in the generation of LTB4 results in a change in the pro-inflammatory status to the anti-inflammatory pathway and enables the removal of neutrophils from the site of inflammation and induces resolution of inflammation [53,54]. This subtle but much needed redirection of the metabolism of AA from forming PGE2/LTB4 to LXA4 to induce resolution of inflammation is critical for resolution of inflammation that depends on the biphasic release of AA from the cell membrane phospholipid pool. It is envisaged that there are two distinct phases of AA-the first phase of release of AA as a result of activation of iPLA2 is utilized for the synthesis of PGE2 whereas the second phase of release of AA due to the activation of cPLA2 and sPLA2 is redirected to form LXA4 (see Figure 3). This unique ability of the cell to regulate the activation of various forms of PLA2s to release specific pools of AA to form PGE2/LTB4 and LXA4 in a very specific and defined fashion to induce the much needed inflammation and subsequently to quell it and lead to resolution of inflammation suggests the unique way cell metabolic processes are tuned as per the necessity of the situation. 

## 7. PGE1 and LXA4 Have Similar Actions

In this context, it is noteworthy that PGE1, derived from DGLA, and LXA4, derived from AA, have similar actions as shown in Table 2. As already discussed above, though EPA is the precursor of 3 series PGs and TXs and 5 series LTs these are less pro-inflammatory compared to 2 series PGs and TXs and 4 series LTs derived from AA. The fact that both pro- and anti-inflammatory metabolites are derived from the same precursor is interesting implying that there is a very well-coordinated and specific functional significance for their formation and suggests how a delicate balance need to be maintained between pro- and anti-inflammatory molecules to restore homeostasis. Any perturbation in their formation and action may lead to failure of resolution of inflammation that results in continuation of the inflammatory process and persistence of the disease.

PGE1 is an anti-inflammatory, vasodilator, platelet anti-aggregator, and cytoprotective and genoprotective molecule and its actions are remarkably like those of LXA4 (see Table 2) but presumably less potent compared to LXA4. It is not known whether PGE1 enhances the formation of LXA4 like PGE2. Previously, we showed that alloxan and streptozotocin-induced inhibition of LXA4 synthesis and secretion by rat insulinoma cells is restored to near normal by GLA (the precursor of DGLA), AA, EPA and DHA (see Figure 4). These results imply that supplementation of GLA, EPA and DHA displace AA from the cell membrane and this displaced AA is converted to LXA4. This may explain some, if not all, of the anti-inflammatory actions shown by GLA, EPA and DHA in vitro and in vivo. This may be in addition to the anti-inflammatory products formed from GLA, EPA and DHA. Furthermore, since the amounts of LXA4 formed in the presence of GLA, EPA and DHA are less compared to LXA4 formed from AA this may account for the limited anti-inflammatory action shown by GLA, EPA and DHA compared to AA. Hence, it is suggested that whenever studies are performed with GLA, EPA and DHA, it is worthwhile to measure LXA4 (in addition to measuring PGs, LTs and TXs and resolvins, protectins and maresins) and compare the amount(s) formed to the degree of anti-inflammatory action noted. 

## 8. Bioactive Lipids and Immune Response

One perceptible feature of aging is a gradual but definite decrease in the resistance to various infections such as COVID-19, influenza, reactivation of tuberculosis and other dormant diseases and cancer. To fight bacterial, viral, and fungal infections and to eliminate cancer cells, the immune system needs to recognize them promptly, adequately and mount the required immune response. In this context, the role of bioactive lipids received much less attention, and more emphasis has been paid to the role of cytokines, and T and B cells. However, it is noteworthy that bioactive lipids can regulate the function of T, B and other immunocytes including macrophages and production and action of various cytokines.

## 9. Anti-Microbial Action 

Studies showed that LA, GLA, DGLA, AA, ALA, EPA and DHA can inactivate both Gram-positive and Gram-negative bacteria, fungi, and enveloped viruses, including influenza and HIV [55,56,57,58,59,60,61,62]. Of all, AA seems to be the most potent. These fatty acids can (i) disrupt cell membrane and induce leakage of their cellular contents resulting in their inactivation; (ii) inhibit microbial respiratory activity; (iii) alter their ability to transport amino acids; and (iv) uncouple oxidative phosphorylation, events that render microbes inactive and non-infective. Alveolar macrophages, leukocytes, T and B cells, NK cells and other immunocytes release AA and other unsaturated fatty acids that can inactivate microbes including viruses SARS-CoV-2, SARS and MERS, implying that appropriate release of unsaturated fatty acids could be one of the fundamental mechanisms by which immunocytes prevent or protect mammals from various microbial infections. Hence, it is suggested that deficiency of AA and other unsaturated fatty acids may render a person susceptible to various infections. With advancing age a decrease or even a deficiency of GLA, DGLA, AA, EPA and DHA may occur due to decrease in the activity of desaturases that may lead to increased susceptibility to infections in the elderly especially to influenza, SARS-CoV-2, SARS, and MERS [58,59,60,61,62,63]. 

## 10. Phospholipase A2 (PLA2) Has Antimicrobial Action

The enzyme phospholipase A2 (PLA2) that induces the release of various PUFAs from the cell membrane lipid pool also has antimicrobial action. Secreted PLA2-IIA is present in human biological fluids in sufficient amounts to induce the release of AA and other unsaturated fatty acids to inactivate several microbes. Secretary PLA2-IIA acts preferentially on phosphatidylglycerol, the main phospholipid component of bacterial membranes at exceptionally low concentrations. In contrast, relatively high concentrations of sPLA2-IIA are required to act on the host cell membranes and surfactant which are predominantly composed of phosphatidylcholine that is a poor substrate for sPLA2-IIA. Transgenic mice over-expressing human sPLA2-IIA are resistant to *Staphylococcus aureus, Escherichia coli,* and *Bacillus anthracis* infections. Intranasal administration of recombinant sPLA2-IIA protects mice from pulmonary anthrax. These results suggest that sPLA2-IIA and AA, EPA and DHA could be administered intranasally to protect against various microbial infections including SARS-CoV-2, SARS and MERS that infect humans through nasal cavity due to the presence of ACE2, the receptor for these coronaviruses [64,65,66]. 

## 11. PUFAs Mediate the Microbicidal Action of Macrophages 

Inhaled staphylococci are eliminated by alveolar macrophages by secreting AA and other PUFAs. Extracellular fluid in the lungs contains significant amounts of AA and other PUFAs that have microbicidal action and thus, compensate for the poor chemotactic and phagocytic capacity and limited intracellular antimicrobial action of alveolar macrophages. AA and other PUFAs are secreted into the extracellular alveolar fluid by macrophages and surrounding cells including leukocytes and T and B lymphocytes [59,60,61,62,67,68,69,70]. Macrophages, T and NK cells and cytotoxic lymphocytes release GLA, AA and other PUFAs to bring about their microbicidal and tumoricidal actions [67,68,69,70,71,72]. This interaction/crosstalk among alveolar lining material, alveolar macrophages, tumor cells and tumor microenvironment include tumor infiltrating macrophages, NK cells, T cells and other immunocytes. It is likely that immunocytes and tumor milieu secrete AA and other PUFAs (including LXA4) not only restrain tumor cell growth but also induce tumor cell apoptosis. One potential possibility is that normal cells surrounding the tumor cells secrete AA, EPA, DHA and LXA4 to inhibit tumor cell growth. It is also likely that tumor cells may uptake the secreted AA and other PUFAs and convert them to PGE2/PGE3 and LTs that have immunosuppressive actions. Thus, there is constantly an intense competition between normal and tumor cells for the uptake of GLA/DGLA/AA/EPA/DHA (by the normal cells). If normal cells are able to take up these fatty acids and produce significant amounts of lipoxins resolvins, protectins and maresins and generate NO and ROS, then tumor cell apoptosis occurs, whereas if tumor cells are able to utilize the released AA to produce PGE2/PGE3/LTs then tumor cells will induce host immunosuppression and proliferate. Aging associated decreased production of GLA/AA/EPA/DHA by cells could lead to decreased production of anti-tumor lipids and thus enhances the risk of cancer development. This is supported by the observation that aged mice produce reduced amounts of anti-inflammatory resolvins, protectins and maresins and LXA4 and higher concentrations of pro-inflammatory LTs when challenged with zymosan [73] (see Figure 5 and Figure 6).

## 12. Interaction between Microbes and Host Cells/Tissues

Gut microbiota plays a major role in several local and systemic diseases. Intestinal dysbiosis may underlie in the development and progression of insulin resistance, type 2 diabetes mellitus, hypertension, cancer, and cardiovascular diseases [74,75,76,77,78,79], diseases that are common with aging. The beneficial actions of gut microbiota could be due to their ability to secrete small chain fatty acids such as acetate, butyrate, and propionate [77,78,79]. These short chain fatty acids have anti-inflammatory actions; reduce splenic effector memory T cells and splenic T_H_17 cells; decrease cardiac immune cell infiltration; abrogate regulatory T cell dependent angiotensin-II-induced hypertension; decrease gut permeability and thus, prevent endotoxemia; act via G-protein coupled receptors or histone deacetylase to act on brain via direct humoral actions, and indirect hormonal and immune pathways to regulate gut-brain-immune axis; and regulate cognitive function and emotions; regulate and communicate via vagal action [77,78,79]. 

However, some of the gut microbiota may also have harmful actions. Obviously, a competition exists between the invading microorganisms and/or harmful gut microbiota and the host tissue/cells for the dietary EFAs. LA and ALA are utilized by the intestinal epithelial cells and T cells/NK cells to form GLA/DGLA/AA/EPA/DHA for their own benefit especially, to inactivate harmful microbes as these lipids have anti-microbial actions [55,56,57,58,59]. This could be one mechanism by which intestinal epithelial cells are able to ward off microbial infections. In addition, intestinal epithelial cells and local T cells/NK cells could use GLA/DGLA/AA/EPA/DHA to form beneficial metabolites such as PGE1, PGI2, NO, CO, H2S, LXA4, resolvins, protectins and maresins to maintain gut epithelial barrier function, maintain gut integrity, protect the gut from various infections, maintain adequate blood supply (since PGE1, PGI2 and LXA4 have vasodilator, prevent platelet aggregation and suppress inappropriate leukocytes and macrophage activation and inhibit excess production of pro-inflammatory cytokines) and nutrition to the gut [80,81,82]. In contrast to this, gut is constantly exposed to pathogenic microbes that alter the inflammatory signaling pathways by reducing degradation of IkB-α and IkBβ that reduces the activation of NF-kappaB in response to exposed microbes [83] that is needed for normal immune response. Some bacteria increase COX-2 expression in macrophages infected with colon cancer *E. coli* compared with macrophages infected with commensal indicating that tumor- infiltrating bacteria have the ability to enhance the expression of COX-2 that, in turn, will enable the tumor cells and tumor infiltrating macrophages to enhance their PGE2, a immunosuppressor [84]. Even the immunosuppression observed in patients with Mycobacterium could be due to local excess production of PGE2 [85]. It is noteworthy that COX-2 induction by CMV (cytomegalovirus) leads to an increase in the production of PGE2 in the host cells that significantly increased CMV proliferation [86]. These results emphasize the importance of gut microbiota in the regulation of production of pro-inflammatory and anti-inflammatory cytokines and bioactive lipids and their role in the maintenance of balance among them to ensure gut homeostasis [87,88]. Thus, it is suggested that beneficial gut microbes enhance the production of GLA/DGL/AA/EPA/DHA that will be utilized for forming adequate amounts of LXA4, resolvins, protectins and maresins to suppress inappropriate inflammation whereas harmful gut bacteria convert these lipids (GLA/DGLA/AA/EPA/DHA) to form pro-inflammatory PGE2 and LTs. This is in addition to their (gut microbiota) ability to influence the production of cytokines. It is not yet known whether SCFAs can alter/influence the formation of PGE2, LTs, LXA4, resolvins, protectins, and maresins. However, it is a strong possibility. Some preliminary evidence suggests that SCFAs can suppress PGE2 production [88] and thus, may indirectly promote LXA4 (including resolvins, protectins and maresins) formation since under physiological conditions a balance is maintained between pro- and anti-inflammatory BALs.

These results imply that a close interaction exists between gut microbiota and BALs that ultimately have a regulatory role in human health and disease. AA, EPA and DHA are known to enhance the proliferation of useful gut microbiota [89,90,91,92,93] that may result in a decrease in the production of pro-inflammatory cytokines and an increase in the elaboration of SCFAs by the gut microbiota. It is not known whether gut microbiota can metabolize PUFAs to SCFAs. However, this is a distinct possibility [93] that needs to be investigated. 

## 13. Immunoregulatory Actions of Bioactive Lipids

Macrophages need to interact with tissue-resident memory CD8^+^ T cells to be effective in their action and function and to sense pathogens and protect the barrier function of tissues [94]. This interaction between macrophages and tissue resident T cells is somewhat like the crosstalk seen between alveolar lining material and alveolar macrophages [95] that is needed to bring about their (macrophages) antimicrobial action. This interaction between epithelial cells and macrophages and between macrophages and other T cells is dependent on the ability of tissue resident memory T cells to uptake fatty acid(s) not only for their survival but also for their specified function. This uptake of the fatty acid(s) is regulated by the type of fatty acid-binding protein (FABP) expressed by the T cells. Tissue resident memory T cells show varying patterns of FABP isoform usage that is decided by tissue-derived factors. Tissue resident memory T cells including macrophages and other immunocytes, modify their FABP expression depending on the tissue in which they are located or relocated, suggesting that immunocytes including tissue resident memory T cells change their expression of FABP based on local conditions [96]. These results emphasize the crosstalk between resident T cells and other immunocytes (including macrophages) and the local tissues/cells in which they (immunocytes) are resident for which (for the cross talk to occur) they (immunocytes) need a specific type of bioactive lipid. This implies that specific type of fatty acids and their metabolites are needed not only to help survive but also to bring about their (T cells and macrophages) action(s) that need to be tailored to their location as dictated by the local milieu. This indicates that specific fatty acids (especially BALs) are needed for proper communication between T cells/macrophages and other immunocytes and the resident tissues/cells and their (macrophages and T cells) ability to inactivate the invading microbes including SARS-CoV-2. It is possible that macrophages/immunocytes/T cells will be able to survive in their host tissue only if their fatty acid content (most probably of their cell membranes) are complementary to each other. This implies that the cell membrane fatty acid pattern/content of macrophages/T cells and other immunocytes is like that of the host cell membrane fatty acid content/pattern. Such an analogy between fatty acid content/pattern of the tissue resident macrophages and host cells/tissues is meant to recognize the invading microbes that are likely to contain a different pattern or type of fatty acids. Thus, cell membrane fatty acid content/pattern is used to recognize self or non-self and mount an immune attack accordingly.

## 14. M1 and M2 Macrophages and Bioactive Lipids

M1 macrophages are pro-inflammatory in nature whereas M2 have anti-inflammatory properties. PGE2 and LTs facilitate the formation of M1 macrophages whereas anti-inflammatory cytokines and LXA4, resolvins, protectins and maresins enhance generation of M2 macrophages [58,97,98,99,100,101,102]. The switch over from M1 to M2 macrophages is needed for the inflammation process to subside and resolution to occur and homeostasis restored. For this process to occur in an orderly fashion, availability of GLA/DGLA/AA/EPA/DHA and their metabolites and the activities of desaturase, COX LOX, PGDH, and sEH (soluble epoxy hydroxylase) enzymes, an appropriate balance between pro- and anti-inflammatory cytokines, generation of ROS, NO, CO, H_2_S and sufficient activity of various anti-oxidants, and appropriate communication among leukocytes, macrophages, T cells, NK cells and CT cells and host cells/tissues and the invading microbes is essential. 

In this context, the crosstalk between pro-inflammatory cytokines and EFA metabolism needs special attention. TNF-α and IL-6 block the activities of desaturases and so the formation of GLA, DGLA and AA and EPA and DHA from their precursors LA and ALA respectively will be defective [22]. In view of this, formation of LXA4 from AA, resolvins from EPA and DHA and protectins and maresins from DHA will be inadequate leading to non-healing of the wound or resolution of inflammation. This is supported by the observation that plasma levels of GLA, DGLA, AA, EPA and DHA are low in those with rheumatoid arthritis, lupus, and sepsis [103,104,105,106] (see Table 3). Aging is associated with decreased activities of desaturases and low plasma concentrations of LXA4 [107,108,109,110]. Based on these results, it is imperative to suggest that administration of GLA/DGLA/AA/EPA/DHA and their anti-inflammatory metabolites LXA4, resolvins, protectins and maresins or their stable and orally active synthetic molecules could be of benefit in inflammatory conditions and potentially to arrest aging process. 

## 15. PGE2 and LXA4 Interact to Induce Resolution of Inflammation

In general, it is believed that PGE2 is a pro-inflammatory molecule. However, several studies suggest that PGE2 is essential to initiate or trigger anti-inflammatory process [111,112,113]. This paradoxical yet beneficial property of PGE2 can be attributed to its ability to bind to different types of PGE2 receptors (EPs). At low concentrations, PGE2 binds to the high-affinity EP4 receptor and enhance IL-23 production, whereas high PGE2 concentrations bind to EP2 receptor and suppress IL-23 release [51,52]. This differential binding of PGE2 explains its dual pro and anti-inflammatory actions [114,115,116,117]. 

In addition, when the concentrations of PGE2 reach a peak, it triggers the generation of LXA4 and suppresses LTB4 production by acting on 5- and 15-lipoxygenase enzymes that enables pro-inflammatory status to be switched over to anti-inflammatory pathway (see Figure 3A,B). This switchover to anti-inflammatory pathway (LXA4) from the pro-inflammatory (LTB4) stage leads to removal of neutrophils and other debris from the site of injury and results in resolution of inflammation [116,117]. In this process of shifting from the pro-inflammatory pathway to the anti-inflammatory status there is a critical role for cytokines as shown in Figure 3B. IL-1β, IL-4 and IL-10 seem to trigger the generation of anti-inflammatory LXA4, PGJ2, and PGD2. Though the exact mechanism by which the cell can switch the metabolism of AA from PGE2 to LXA4 is not clear, but at least, in part, it depends on the source of AA. It has been suggested that there is a biphasic release of AA from the cell membrane lipid pool. The first pulse of release of AA is due to the activation of iPLA2 that is utilized to form predominantly PGE2, whereas the second phase of release of AA is as a result of the activation of cPLA2 and sPLA2 that is directed to form LXA4 (see Figure 3B). Thus, factors that regulate the activation of different forms of PLA2 are uniquely placed to regulate inflammation. It is not known whether PGE1 derived from DGLA, the precursor of AA, also has actions like PGE2 in triggering the formation of LXA4. 

PGH1 is the cyclo-oxygenase metabolite of DGLA and the precursor of PGs of 1 series that have anti-inflammatory actions. PGH1 is an activator of the pro-inflammatory PGD2 receptor CRTH2 {chemoattractant receptor homologous molecule expressed on T helper type 2 (Th2) cells}. PGH1 mediates migration of Th2 lymphocytes, shape change of eosinophils, and their adhesion to human pulmonary microvascular endothelial cells indicating that it (PGH1) has pro-inflammatory actions [118] but certainly much less potent compared to PGH2 and PGE2 and LTs (formed from AA) (see Figure 3C). Hence, it can be deduced that PGE1 (unlike PGE2) is unable to trigger significant inflammation resolution events. Similarly, even EPA may not have the ability institute adequate inflammation and resolve inflammation since PGE3 and LTs formed from EPA are not potent pro-inflammatory molecules compared to PGE2 and LTs derived from AA. PGE1 is an anti-inflammatory, vasodilator, and platelet anti-aggregator and has many actions like LXA4 (see Table 2) though it is apparently much less potent compared to LXA4. 

## 16. PGE2 Is Needed for Tissue Regeneration 

Once inflammation subsides, healing of the wound and tissue regeneration is needed to restore homeostasis. Enhanced concentrations of PGE2 at the site of inflammation and elsewhere is needed not only to trigger LXA4 synthesis from AA but also to facilitate tissue regeneration. Inhibition of 15-PGDH (15-prostaglandin dehydrogenase, a prostaglandin degrading enzyme) produced a two-fold increase in PGE2 concentrations in many tissues including the bone marrow, colon, and liver and enhanced hematopoietic capacity. It is noteworthy that 15-PGDH deficient animals showed a rapid liver regeneration after partial hepatectomy and enhanced recovery of neutrophils, platelets, and erythrocytes [119]. This is supported by other studies which showed that PGE2 is needed to promote hematopoiesis [120,121,122,123]. Based on these results, it is likely that administration of AA, the precursor of both PGE2 and LXA4, may result in the formation of both LXA4 for inflammation resolution and PGE2 to enhance tissue regeneration.

## 17. LXA4 Is the Mediator of Beneficial Action of MSCs (Mesenchymal Stem Cells)

Acute lung injury and diabetic renal damage can be resolved by MSCs due to their ability to secrete LXA4 [124,125,126]. Hence, it is possible that the beneficial action of MSCs in COVID-19 can be attributed to their ability to secrete LXA4 [63]. These results imply that both PGE2 and LXA4, derived from AA, have beneficial actions depending on the context. In general, the delicate balance between PGE2 and LXA4 is needed to trigger inflammation and to resolve inflammation and to restore homeostasis. Inhibition of PGE2 synthesis that can trigger LXA4 formation from AA is known to augment antiviral immunity by inducing type I interferon production and apoptosis of macrophages [23,59,60,61,62,63,124,125,126,127,128,129].

AA is metabolized by cytochrome P450 to epoxyeicosatrienoic acids (EETs) that have anti-inflammatory action. EETs are degraded by soluble epoxy hydrolase (sEH). LXA4 inhibits the activity of sEH and thus, may bring about some of the beneficial actions of EETs [130]. ALA (and possibly, other fatty acids such as AA/EPA/DHA) inhibits the activity of sEH [131], suggesting that other bioactive lipids may possess similar action on sEH, yet another mechanism by which they (bioactive lipids) are able to suppress inflammation in several diseases (see Figure 3B).

Thus, BALs are involved in the regulation of inflammation and immune response, wound healing, regeneration of tissues and restoration of homeostasis, events that are compromised with advancing age. With increasing age, the activities of desaturases decreases accompanied by an increase in COX-2 activity, alteration in LOX enzymes, increase in the production of IL-6 and TNF-α and a decrease in IL-4 and IL-10, anti-inflammatory cytokines, an increase in PGE2 and LTXB4 formation with a concomitant decrease in LXA4 (and possibly, resolvins, protectins and maresins) resulting in a tilt in the balance more towards pro-inflammatory events (see Figure 3B, Figure 5 and Figure 6). These events may lead to an increase in insulin resistance, peripheral vascular resistance resulting in the development of hypertension, metabolic syndrome and coronary heart disease and CNS disorders such as Alzheimer’s disease, and other chronic inflammatory diseases seen with increasing age. Hence, strategies designed to augment the action of desaturases, suppress COX-2 and regulate LOX enzymes and restore the balance among PGE2, LTB4 and LXA4 (and resolvins, protectins and maresins) will lead to suppression of inappropriate inflammation and prevention of age-related diseases. It is proposed that supplementation of preferably AA (and limited EPA and DHA) could lead to formation of LXA4 and decrease in PGE2 production and inhibition of IL-6 and TNF-α and restoration of homoeostasis and may prevent or ameliorate all aging associated diseases. 

## 18. Low-Grade Systemic Inflammation Occurs in Aging and Aging Associated Disorders

Aging is a low grade systemic inflammatory condition as evidenced by the observation that pro-inflammatory cytokines are increased with advancing age [132,133,134,135,136]. Chronic, progressive low-grade systemic inflammation is seen in aging associated conditions such as obesity, insulin resistance, type 2 diabetes mellitus, hypertension, metabolic syndrome, and cancer. Knockout of the nfkb1 subunit of the transcription factor NF-κB that induces low grade systemic inflammation causes premature aging in mice associated with reduced regeneration in liver and gut like reduced or defective healing seen with advanced age. In addition, nfkb1(−/−) fibroblasts show aggravated cell senescence that is associated with increased activity of NF-κB and COX-2 and ROS generation. Increased activity of NF-kB and augmented expression of COX-2 instigates oxidative stress and produces telomere dysfunction [134,135]. These results suggest that inflammation is an important factor involved in aging, telomere shortening, and decreased regenerative capacity. Since aging increases the incidence of insulin resistance, obesity, hypertension, type 2 diabetes mellitus and cancer, it is reasonable to suggest that endothelial dysfunction, atherosclerosis, diabetes mellitus, hypertension, coronary heart disease and cancer are associated with an imbalance in the concentrations of pro- and anti-inflammatory bioactive lipids such that a deficiency of anti-inflammatory lipoxins, resolvins, protectins and maresins and NO (nitric oxide) and a relative increase of PGE2, PGF2α concentration occurs. 

In addition, BALs are involved in several cellular functions and biological processes that are altered in aging. Some of them include: (i) actions of ion channels including Piezo1 and Piezo 2 and voltage gated ion channels such as TrpV1 and thus, participate in cell mitotic process, cell signaling, cell cycle progression, as well as cell volume regulation; (ii) alter cell membrane composition that can influence cell membrane fluidity and expression of several receptors and their expression and the structure and composition of intermediate filaments and their multiple binding partners and thus, regulate both cellular mechanics and gene regulation, (iii) regulate mitochondrial processes; (iv) telomerase activity; and (v) G-protein–mediated signal transduction [23,39] in addition to their ability to modulate inflammation, immune response and stem cell biology [23,63,134,135,136,137].

## 19. Regulatory Action of Bioactive Lipids on Sirtuins

Sirtuins are a class of proteins that have either mono-ADP-ribosyltransferase, or deacylase activity, including deacetylase, desuccinylase, demalonylase, demyristoylase and depalmitoylase activity and have been implicated to have a role in aging, transcription, apoptosis, inflammation and stress resistance. The sirtuin-mediated deacetylation reaction couples lysine deacetylation to NAD hydrolysis that yields O-acetyl-ADP-ribose and nicotinamide. The dependence of sirtuins on NAD^+^ links their enzymatic activity to the energy status of the cell that depends on the cellular NAD^+^-NADH ratio. Sirtuins deacetylate histones for which Zn^2+^ is needed as a cofactor. Sirtuin activity is inhibited by nicotinamide. In view of its actions, sirtuins have a critical role in aging, DNA repair and function as tumor suppressors. 

Some free fatty acids have been shown to stimulate SIRT1 through protein kinase A-dependent activation of SIRT1-PGC1α complex and thus, increase rates of fatty acid oxidation and prevent lipid dysregulation associated with aging [138]. SIRT1 activation is known to prevent aging. Of all the SIRTs, SIRT6 is critical for intermediary metabolism and genomic stability [139]. SIRT6-deficient mice have shortened lifespan, defects in DNA repair and a high incidence of cancer due to oncogene activation [139,140]. In contrast, SIRT6 overexpression lowers LDL and triglyceride level, improves glucose tolerance [141], and increases lifespan of mice [142] in addition to its anti-inflammatory effects at the transcriptional level [143]. 

GLA, DGLA, AA, EPA and DHA and their metabolites influence the activity of SIRT6 and other SIRTs and thus, bring about their actions on metabolism, inflammation, and genome maintenance. GLA, DGLA, AA, EPA and DHA and prostaglandin E2 (PGE2), lipoxin A4 (LXA4) (pro- and anti-inflammatory metabolites of AA respectively) activate/suppress various SIRTs (SIRt1 SIRT2, SIRT3, SIRT4, SIRT5, SIRT6), PPAR-γ, PARP, p53, SREBP1, intracellular cAMP content, PKA activity and peroxisome proliferator-activated receptor γ coactivator 1-α (PGC1-α). Changes in the metabolism of DGLA, AA, EPA and DHA as a result of altered activities of desaturases, COX-2 and 5-, 12-, 15-LOX in various tissues may play a critical role in determining age and development of several aging associated diseases. Thus, methods designed to maintain homeostasis of GLA, DGLA, AA, EPA, DHA, PGE2, LXA4 may arrest aging process and associated metabolic abnormalities. Measurement of the concentrations of various bioactive lipids and the genes/enzymes needed for their formation could be employed to gauge the status of aging process and as targets to develop suitable remedial measures to prevent/arrest/postpone the aging process itself. 

Cytochrome P450 (CYP) epoxygenases convert AA to epoxyeicosatrienoic acids (EETs) (see Figure 2), which protect the renal system from ischemia/reperfusion (I/R)-induced acute kidney injury by acting on SIRT1. Exogenous 11,12-EET addition suppresses I/R-induced apoptosis through SIRT1-FoxO3a signaling activation. These results suggest that CYP2J2-produced EETs activate the SIRT1-FoxO3a signaling pathway to bring about their beneficial actions [144]. 15-HETE promotes the transcription and translation of SIRT1 to increase viability of pulmonary arterial smooth muscle cells by promoting the expression of Bcl-2 and Bcl-xL [145]. 15-HETE is crucial for the protection of PASMCs (pulmonary arterial smooth muscle cells) against cell death, and the SIRT1 pathway may provide a new strategy for pulmonary artery hypertension therapy, in part, by acting on nitric oxide, PI3/AKT and ERK1 and ERK2 pathways [146,147,148] to produce its beneficial actions indicating that a close interaction exists among BALs, SIRTs, PI3/AKT, and ERKs. EPA and DHA, PGs and resolvins may have similar action on SIRTs [149,150,151,152,153,154], see Figure 1 and Figure 7). In addition, BALs can modulate the properties of ion channels, mitochondrial processes, and G-protein-mediated signals that may also explain their involvement in several diseases [23]. 

## 20. Bioactive Lipids in Age-Associated Diseases

Accumulation of abdominal fat, decrease in muscle and skeletal mass (osteoporosis), development of insulin resistance, type 2 diabetes mellitus, hypertension, cancer, coronary heart disease (CHD), atherosclerosis, Alzheimer’s disease, depression, spinal disc prolapse, and osteoarthritis are common with aging. All these diseases are associated with low-grade systemic inflammation in which BALs play a role. BALs by virtue of their ability to alter cell membrane fluidity, influence ion channels, action on G protein coupled receptors, regulation of inflammation, immune response, stem cell biology, telomerase activity, mitochondrial processes, and cytoskeletal system could play a role in these diseases. Based on the preceding discussion, it is opined that measuring plasma/tissue concentrations of GLA, DGLA, AA, EPA, DPA, DHA, PGs, LTs, TXs, HETEs, EETs, sEH, lipoxins, resolvins, protectins and maresins and anti-inflammatory cytokines, NO, H_2_S, and CO may offer clues to their potential role in these age-related diseases and offer clues as to the potential therapeutic strategies. 

Despite all the evidence and arguments presented, it is hard to believe how one set of molecules (namely BALs) may have a role in such a wide variety of conditions. One possibility is that it is their local tissue concentrations and actions are of specific importance. Abnormalities in the concentrations of various BALs including the activities of desaturases, COX, LOX, PGDH P450 enzymes and the expression of their respective receptors in specific tissues determine their involvement in various diseases. For instance, altered BALs system in the vascular endothelial cells may cause hypertension, pancreatic β cells to diabetes mellitus, adipose tissue to obesity, skeletal muscles to sarcopenia, osteoclasts and osteoblasts to osteoporosis, coronary vascular endothelial cells to atherosclerosis, neuronal cells of the brain to Alzheimer’s disease and depression, and in specific cells to relevant cancer(s). Previous studies showed that tumor cells are particularly deficient in AA due to low activity of desaturases and similarly, there are alterations in the plasma concentrations of various unsaturated fatty acids in other diseases as well (see Table 4). These results imply that administration of various PUFAs and/or lipoxins, resolvins, protectins and maresins in appropriate amounts and in a timely manner may prevent or even reverse these diseases. Since lipoxins, resolvins, protectins and maresins are unstable and have short half-life, their precursors such as GLA/DGLA/AA/EPA/DPA/DHA may be administered. 

## 21. AA in Aging

It is evident from the preceding discussion to propose that administration of the deficient BALs may form a new therapeutic approach both to prevent and arrest not only the aging process but also diseases associated with aging. This is supported by the observation that in many diseases (especially in aging associated diseases such as hypertension, type 2 diabetes mellitus, and coronary heart disease) there is indeed a decrease in the plasma concentrations of GLA. DGLA, AA, EPA and DHA, the precursors of both pro- and anti-inflammatory eicosanoids (see Table 3, Table 4 and Table 5) [117,155,156]. The activities of desaturases decrease whereas that of COX-2 increases and alterations in the activities of 5-, 12- and 15-LOX have also been described with aging [23,136,157,158,159,160,161,162,163,164,165,166,167,168] that may result in changes (specifically decrease) in the concentrations of GLA, DGLA, AA, EPA and DHA leading to increased production of PGE2, LB4/LTE4 and decreased formation of LXA4, resolvins, protectins and maresins synthesis tilting the balance more towards pro-inflammatory status. Similar imbalance in the cytokines more towards pro-inflammatory status with an increase in IL-6, TNF-α and HMGB-1 and a decrease of anti-inflammatory IL-4, IL-10 and IL-12 occurs with aging. This increase in pro-inflammatory milieu that could form the basis of several aging associated diseases [23,108,109,155,157,158,159,160,161,162,163,164,165,166,167,168]. It is noteworthy that exercise and dietary restriction enhance the activities of desaturases and increase the formation of LXA4 (and possibly, that of resolvins, protectins and maresins) [23,136,157,161,169,170]. Thus, two well-known interventions that can postpone aging—dietary restriction and exercise—induce anti-inflammatory state, lending support to the proposal that aging is a low-grade systemic inflammatory condition with corresponding changes in the BALs and cytokines (see Figure 5, Figure 6 and Figure 7).

## 22. AA in *C elegans* and Life Span

Since it is not possible to study all the desired studies in humans and sometimes in experimental animals, model organisms are employed for this purpose. *Caenorhabditis elegans (C. elegans*) is one such organism employed for this purpose since it contains many genes that are also present in humans with similar properties. *C. elegans* when fed with high-glucose diet shows shortened lifespan and reduced LA and AA concentrations [171,172]. Importantly, ω-6 PUFAs attenuated the short lifespan due to high-glucose-feeding in C. elegans [171,172]. *C. elegans* mutants that develop PUFA deficiency especially of AA showed developmental and behavioral abnormalities such as slow growth, less spontaneous movement, altered body shape, abnormalities in timing of the circadian rhythm, defecation cycle and produced fewer progeny compared to the wild type. All these abnormalities could be ameliorated by supplementing these *C. elegans* with diet rich in GLA or AA or EPA [171,172]. These and several other studies attest to the fact that BALs are needed for various functions such as growth and development (especially of brain), neurotransmission, behavior, circadian rhythm, reproduction, immune response, to prevent high-glucose-induced life-span reduction, regulation of obesity, and several other features that are essential for normal life span [171,172,173,174], implying that one of the critical functions of BALs could be to regulate aging and aging associated disorders including obesity, insulin resistance, type 2 diabetes mellitus, metabolic syndrome, Alzheimer’s disease, depression, and cancer [23,58,117,155,175,176,177]. These results are in tune with the observation that patients with type 2 DM, hypertension, coronary heart disease, cancer, lupus, RA and pneumonia have altered PUFA metabolism [155] (see Table 3, Table 4 and Table 5).

## 23. Alzheimer’s Disease and BALs

Increasing age is a major risk factor for Alzheimer’s disease (AD) and the most common form of dementia [178]. Aging is known to be associated with increasing incidence of hypertension and type 2 diabetes mellitus that are believed to enhance the risk of Alzheimer’s disease [179,180,181]. In view of the close association between insulin resistance and decreased cerebral glucose metabolism with Alzheimer’s disease, it has been argued that Alzheimer’s could be considered as a type 3 diabetes mellitus [182]. Such a purported relationship between T2DM and AD led to the use of intranasal insulin in its (AD) treatment resulted in controversial results [183]. Furthermore, changes in lipid metabolism in brain regions that play a significant role in cognitive function [184,185] which is reflected in the periphery [181] and with age have been described [186]. In a recent study that was coined to evaluate age-related changes in the lipidome of the rat amygdala obtained from young (3 months) and old (20 months) males of the Sprague-Dawley rats, significant changes in the levels of four main lipid classes: glycerolipids, glycerophospholipids, sphingolipids and sterol lipids was described [184]. It was reported that a significant level changes in those of phosphatidic acid, diacylglycerol, sphingomyelin, and ceramide that are involved in lipid signaling and affect amygdala neuronal activity [184]. Since amygdala is a crucial brain region for cognitive functions, these results imply that there is a significant role for lipid molecules in age-associated memory and AD. Similar complex association of sphingolipids and PUFAs with AD based on age and gender has also been described [187].

In this context, it is noteworthy that significant changes in the phospholipids was reported in the hippocampus due to aging [188]. It was reported that the phospholipid composition of the mitochondrial and microsomal membranes of the human hippocampus from post-mortem tissue of normal subjects aged between 18 and 104 years showed decreases with age of adrenic and AA. Since aging is associated with increased incidence of AD, it is assumed that decrease in AA and possibly, EPA and DHA may be associated with AD. This supposition is supported by the observation that both DHA and AA form an important constituent of neuronal membranes, and in combination with EPA can affect cardiovascular health and modulate inflammatory events and are needed for resolution of inflammation. Furthermore, deficiency of these PUFAs seem to have a role in disorders like schizophrenia and attention deficit hyperactivity disorder (ADHD). Perinatal supplementation of AA and DHA is critical for normal brain growth and development in humans and seem to improve cognition and sensorimotor integration [189,190,191,192,193]. In view of these important functions of AA and DHA, it is possible that they are involved in the pathobiology of AD and Parkinson’s disease. This is supported by the observation that plasma PL (phospholipid), phosphatidylcholine (PC) and plasma phosphatidylethanolamine (PE) levels of EPA, DHA and n−3/n−6 ratio were lower in those with AD [194]. In some of these studies [194] plasma AA levels were found to be high, the exact reason for this is not clear.

The release of various PUFAs from the cell membrane lipid pool depends on the activity of PLA2 (phospholipase A2). This led to the study of PLA2 activity in AD. These results showed elevated levels of cPLA2 immunoreactivity in AD brain that led to the suggestion that an active inflammatory process occurring in AD [195]. Since there are many types of PLA2s (~ 12) and each of them have variable biological activity, substrate specificity, activating factors and localization, deciphering individual actions are difficult if not impossible. Of all, GIVA-PLA2 (group IVA-PLA2) specifically releases AA and is expressed constitutively in neurons, whereas GVIA-PLA2 is expressed by many cell types and brain regions and, in contrast to GIVA-PLA2, does not require Ca^2+^ for its activation. GVIA-PLA2 predominantly induces the release of DHA in the brain [196,197]. Both PLA2 and AA are important for synaptic signaling, long-term potentiation (LTP), learning and memory [198,199,200,201,202]. Inhibition of PLA2 diminished memory especially specific inhibition of GVIA-PLA2 implying a role for AA [203,204,205]. Several types of stimuli that enhance neuronal activity also increase PLA2 activity and AA levels to enhance neuronal activity that, in turn, increase the production of Aβ (amyloid beta). Aβ increases GIVA-PLA2 and AA release implying a feedback regulation among these factors (Aβ, GIVA-PLA2 and AA). The ability of AA to affect neuronal activity and functions may occur by its direct action or indirectly by its various metabolites such as PGs, LTs and HETEs (hydroxyeicosatetraenoic acids) [206,207,208].

Surprisingly, there are very few reports as to the effects of anti-inflammatory metabolite of AA namely LXA4 on neuronal function, synaptic plasticity, and memory. It was reported that formyl-peptide receptor 2 (FPR2), the common receptor for LXA4 and resolvin D1, mRNA expression can be seen in many areas of the brain (brainstem > spinal cord > thalamus/hypothalamus > cerebral neocortex > hippocampus > cerebellum > striatum) with the brainstem and spinal cord containing the highest levels of FPR2 protein. FPR2 seems to be able to enhance the lengths of axons and dendrites. These results suggest a role for FPR2 (indirectly for LXA4 and resolvin D1) in learning and memory, balance and nociception that could be because of FPR2 in mediating AA/LXA4 or DHA/RvD1-induced axonal or dendritic outgrowth [209]. These results are supported by the report that knockdown of ALOX-15 (that is necessary for the formation of resolvin D1 from DHA) inhibited LTP of hippocampo-prefrontal cortex pathway and resulted in an increase in the errors in alternation, in the T-maze test. This suggests that resolvin D1 is needed for LTP at hippocampo-prefrontal cortical synapses and associated spatial working memory performance. Thus, LXA4 and resolvin D1 are necessary for normal neuronal signaling, and cognition [210]. It is likely that even protectins and maresins may have actions like LXA4 and resolvin D1 on cortical synapses and memory performance. These results suggest that anti-inflammatory molecules such as LXA4, resolvins, protectins and maresins have a key role in axonal and dendritic outgrowth, neuronal function, synaptic plasticity, and memory. These results can be interpreted to mean that neuroinflammatory brain disorders and chronic neurodegeneration such as Alzheimer’s disease could be due to a relative deficiency of these anti-inflammatory molecules (LXA4, resolvins, protectins and maresins and their precursors such as AA, EPA and DHA) and an excess of pro-inflammatory cytokines IL-6, TNF-α and PGs, LTs and TXs. Thus, the balance between these pro and anti-inflammatory molecules may ultimately determines the development and progression of AD. In this context, the activities of GIVA-PLA2 (needed for the release of AA) and GVIA-PLA2 (needed for the release of DHA) and their isoenzymes such as iPLA2, sPLA2 and cPLA2 and COX-2 and LOX enzymes determine the final products that are formed from the released AA and DHA in the development of AD and its progression [211,212].

## 24. PLA2, COX-2, LOX Enzymes, Cytokines and Pro- and Anti-Inflammatory Eicosanoids and AD

It is evident from the preceding discussion that the maintenance of the balance between pro- and anti-inflammatory molecules is critical in the prevention and initiation and progression of AD in which BALs and their metabolites have an important role. The critical role played by PLA2s, COX, and LOX enzymes and cytokines in the initiation of inflammation and its resolution has been discussed previously in detail elsewhere [213] and is describe in brief below.

There are three classes of PLA2 that control the release of AA, DHA and other PUFAs. These include calcium independent PLA2 (iPLA2), secretory PLA2 (sPLA2), and cytosolic PLA2 (cPLA2). There are 10 isoenzymes for sPLA2, at least three for cPLA2, and two for iPLA2. In the early phase of inflammation, COX-derived PGs and lipoxygenase-derived LTs initiate exudate formation and inflammatory cell influx triggered by TNF-α and IL-6 and other pro-inflammatory cytokines. It is expected that during the resolution phase of inflammation an increase in LXA4 and other anti-inflammatory lipid metabolites (such as resolvins, protectins, maresins, PGD2 and 15deoxyΔ12-14-PGJ2) will be generated with a simultaneous reduction in PGE2 formation. This implies that there are two waves of release of AA and DHA-the first wave is meant for the generation of PGE2 and LTs and the second wave for the synthesis of LXA4, resolvins, protectins, and maresins. It was reported that type VI iPLA2 protein is the principal isoform expressed at the onset of inflammation, whereas type IIa and V sPLA2 are expressed at the onset of resolution of inflammation. Type IV cPLA2 is not detectable during the early phase of inflammation but increased progressively during resolution, peaking at 72 h that is mirrored by a parallel increase in COX-2 expression. Thus, there is a clear-cut role for different types of PLA2 in distinct and different phases of inflammation. Selective inhibition of cPLA2 results in the reduction of proinflammatory PGE2, LTB4 and IL-1β. Inhibition of types IIa and V sPLA2 not only decreases LXA4 but also results in a reduction in cPLA2 and COX-2 activities. IL-1β can induce the expression of cPLA2 expression suggesting that IL-1 is not only meant for induction of inflammation but also to induce cPLA2 expression to initiate resolution of inflammation. Synthetic glucocorticoid dexamethasone inhibits both cPLA2 and sPLA2 expression, whereas type IV iPLA2 expression is refractory to its action. In addition, activated iPLA2 assists in the conversion of inactive proIL-1β to active IL-1β, which in turn induces cPLA2 expression that is necessary for resolution of inflammation. TNF-α seems to possess direct suppressive action on the synthesis of LXA4, PGD2, and 15-deoxyΔ^12–14^ PGJ2 from cPLA2-induced release of AA/DHA. In contrast, LXA4 suppresses TNF-α-induced production of ILs; promote TNF-α mRNA decay, TNF-α secretion, and leukocyte trafficking; and thus, attenuates inflammation. This interaction among PLA2s, COX-2, LOX, PGD2 and LXA4 in the initiation, maintenance, and resolution of inflammation is critical to induce much needed inflammation when an infection and injury occurs and initiate resolution of inflammation when it is time to heal. When the balance among these factors is altered it could lead to less optimal inflammation and persistence of inflammation as seen in AD (see Figure 3B).

Several biomarkers significantly associated with APOEɛ4 ‘risk’ and ɛ2 ‘protective’ genotypes (versus neutral ɛ3/ɛ3) for AD [214] are likely to be associated with the activities of PLA2s, COX-2, and LOX enzymes and the formation of various pro- and anti-inflammatory bioactive lipids. However, this association has not yet been studied in detail. Based on the preceding discussion, it is reasonable to propose that deficiency of AA and DHA may enhance the risk of AD [215] and their supplementation may be of therapeutic benefit. It appears whenever there is a deficiency of AA/EPA/DHA, an increase in the production of PGE2, LTs and TXs would occur accompanied by a deficiency of LXA4, resolvins, protectins and maresins as observed in type 2 DM, hypertension, and CHD [35,36,117,155] (see Table 5), diseases in which LXA4 deficiency occurs and are commonly associated with AD. Furthermore, it was shown that in such AA deficiency states supplementation of AA enhances LXA4 formation without increasing PGE2 levels [216,217,218]. Hence, it is worthwhile to supplement AA/DHA to those at high risk and having AD to study potential therapeutic benefit of such an intervention. Such a strategy may also be employed to prevent or arrest or reverse some, if not all, of the features of aging.

## 25. Gangliosides, Sphingolipids, Cholesterol, and Plasmalogens and their Relationship to BALs in Ageing

It needs to be mentioned here that even though in the present discussion the focus has been on bioactive lipids (PUFAs, PGs, LTs, TXs, lipoxins, resolvins, protectins and maresins) in aging, there could be a role for other lipids such as gangliosides, sphingolipids, cholesterol, and plasmalogens in aging especially during disease. It is likely that BALs may have the ability to modulate the metabolism of gangliosides, sphingolipids, cholesterol and plasmalogens. This implies that alterations/changes described in the concentrations and/or metabolism of gangliosides, sphingolipids, cholesterol and plasmalogens could be perceived as secondary to the alterations noted in the concentrations and metabolism of BALs. For instance, it is known that the amount of DHA in the PL species is decreased with aging in the retina. Most lipids especially, choline and serine glycerophospholipids have low DHA content that has been attributed to the decreased synthesis of DHA [219]. Similar decreases in the content of AA, 22:4 n−6, and DHA was noted in ethanolamine and serine glycerophospholipids during aging. Similarly, phosphatidate phosphohydrolase and phospholipase D activities have also been found to be altered in the aged brain that may be responsible for the changes in the lipid second messengers diacylglycerol and phosphatidic acid [220]. It has been reported that supplementation of EPA and DHA improves apoB100 metabolism that has been attributed to changes in sphingolipids implying that BALs can alter sphingolipids metabolism. Thus, any changes in the sphingolipids during aging could be secondary to altered BALs’ metabolism or deficiency of BALs [221]. In this context, it is noteworthy that cholesterol interferes with the activity of Δ^6^ and Δ^5^ desaturases that are needed for the conversion of dietary LA and ALA to their respective long-chain metabolites namely AA, EPA and DHA (see Figure 2A) and GLA, DGLA, AA, EPA and DHA decrease formation of cholesterol by inhibiting the activity of HMG-CoA activity that suggests that increase in plasma and tissue cholesterol content are secondary to alterations in the metabolism of EFAs [10,11,12,13]. These results indicate that potential changes of gangliosides, sphingolipids, cholesterol and plasmalogens could be attributed to alterations in BALs’ metabolism. Despite these arguments, certainly more in-depth studies are needed to understand the relationship between BALs and other lipids especially during aging.

## 26. BALs, Oxidative Stress and Longevity of Naked Mole Rat (NMR)

If the hypothesis proposed here that BALs have a role in the aging process it need to explain the low EFA content in their tissue membranes and more pro-oxidative cellular environment seen in the naked mole rats (NMR, *Heterocephalus glaber*) [222,223]. In a shotgun lipidomics study of the NMR, the longest-living rodent known with a maximum lifespan of > 28 years, it was reported that the total phospholipid distribution is similar in the skeletal muscle, heart, and liver tissues compared to mice. Total phospholipid (PL) distribution is similar in tissues of NMR and mice; DHA is only found in phosphatidylcholines (PC), phosphatidylethanolamines (PE) and phosphatidylserines (PS), and DHA is more concentrated in PE than PC. NMR had fewer species of both PC and PE compared to mice. DHA-containing phospholipids represent 27–57% of all phospholipids in mice but only 2–6% in NMR. Vinyl ether-linked phospholipids (plasmalogens) are higher in NMR tissues than in mice. Since NMR has lower level of DHA-containing phospholipids, one would expect lower levels of lipid peroxides. Contrary to this, NMR has lower GSH and GSH/GSSG indicative of poor antioxidant capacity and more pro-oxidative cellular environment, and have almost 10-fold higher levels of lipid peroxides compared to mice. NMR showed higher levels of accrued oxidative damage to lipids (twofold), DNA (approximately two to eight times) and proteins (1.5 to 2-fold) compared to physiologically age-matched mice. It has been argued that high level of plasmalogens may be responsible for enhanced membrane antioxidant protection seen in NMR compared to mice. These results contrast with the general belief that oxidative stress is responsible for aging.

Previously, we showed that alloxan and STZ (streptozotocin)-induced diabetic animals have high levels of oxidative stress (as evidenced by increased levels of lipid peroxides and decreased antioxidants) that were restored to normal in AA, LXA4, resolvin D1 and protectin-treated animals (alloxan + BALs or STZ + BALs) [34,35,36,224,225] suggesting that BALs have antioxidant action. In addition, we observed that lipid peroxides of AA (and possibly, of EPA and DHA) bind to DNA to regulate gene action especially, those of oncogenes (suppress expression of ras, myc and other oncogenes) and thus, inhibit the development of cancer [226]. It was noted that one mechanism by which tumor cells can be eliminated is by enhancing the formation of toxic lipid peroxides compared to normal cells that accounts for the selective elimination of tumor cells by GLA, AA, EPA and DHA [227,228,229,230,231,232,233]. This ability of selective accumulation of toxic lipid peroxides to induce apoptosis/ferroptosis of tumor cells seems to be dependent on the glutathione peroxidase (GPX4) levels in the cells [230,231,232,233]. Thus, if the cellular content of GPX4 is high the tumor cells are drug-resistant and such drug-resistant tumor cells can be rendered drug sensitive by reducing intracellular GPX4 levels and enhancing accumulation of lipid peroxides. The presence of high levels of lipid peroxides and low concentrations of GSH/GSSG may explain as to why NMR are resistant to cancer. Furthermore, we observed that GLA supplementation to those who have been on long-term diphenylhydantoin therapy for epilepsy have fewer micronuclei containing peripheral lymphocytes (an indication of DNA damage) possibly, due to their elimination (by apoptosis) [234,235]. This has been attributed to the ability of GLA (and possibly, other PUFAs such as AA, EPA and DHA) to render them susceptible to apoptosis by enhancing lipid peroxidation [227,228,229,230,231,232,233]. Thus, presence of high levels of lipid peroxides seen in NMR can be one reason for their long life and reduced incidence of cancer. It is likely that presence of nuclear material in the cytoplasm (in the form of micronuclei) may lead to cGAS expression and consequently their apoptosis by the immunocytes that could recognize these micronuclei containing cells as foreign and eliminate them. These observations lend support to the concept that BALs have a significant role in aging that could be attributed to their ability to enhance lipid peroxidation, regulate gene/oncogene expression, induce apoptosis of tumor cells (abnormal cells that possess DNA damage) and serve as second messengers for a variety of cellular processes.

## 27. Conclusions and Therapeutic Implications

It is evident from the preceding discussion that BALs have several actions that emphasize their potential role in the pathobiology of aging and aging-associated diseases/disorders. This implies that administration of BALs may form a new therapeutic approach in the prevention, postponing or even reversing some of the aspects of aging. Many diseases that are common in the elderly such as obesity, sarcopenia, insulin resistance, type 2 diabetes mellitus, hypertension, CHD, Alzheimer’s disease, depression, cancer are all associated with a decrease in the activities of desaturases, enhanced activity of COX-2 and altered expression of LOX enzymes and possibly, decreased activity of 15-PGDH, and a frank deficiency or subclinical deficiency of various co-factors needed for adequate metabolism of EFAs such as vitamins B1, B6, B12, folic acid, and various minerals and trace elements could be the underlying cause for excess of PGE2. LTB4/LTE4, deficiency of LXA4, resolvins, protectins and maresins, and an imbalance in the cytokines (with the balance tilted more towards pro-inflammatory milieu) seen in them [10,11,12,13,14,15,16,17,23,236,237,238,239]. This implies that in order to determine the interventions that need to be instituted in the elderly, one may have to measure the plasma levels of EFAs, their metabolites (including GLA, DGLA, AA, EPA, DHA, LXA4, resolvins, protectins, maresins, PGE2, PGE1, PGJ2, LTB4, LTE4 and the activities of desaturases, COX-2, LOX, 15-PGDH, CYP450 enzymes, various minerals, trace elements and vitamins and accordingly administer the required BALs and other co-factors.

The half-life of PGE2, PGI2, PGJ2, LXA4, resolvins, protectins and maresins is very shot (not more than few seconds to minutes) and are unstable and hence, it is recommended that their precursors GLA, DGLA, AA, EPA and DHA may be considered for administration to prevent aging associated conditions. GLA, DGLA, AA, EPA and DHA have many overlapping actions especially on inflammation and immune response regulation. Of all, administration of AA/EPA/DHA seems to be more reasonable since they form precursors to beneficial metabolites LXA4 (from AA), resolvins (from EPA and DHA), protectins and maresins (from DHA). Both AA and EPA also form precursors to various PGS, LTs and TXs that have pro-inflammatory actions implying that under normal physiological conditions a delicate balance is maintained between their pro- and anti-inflammatory metabolites. It is noteworthy that PGs of 2 series, TXs of 2 series and LTs of 4 series derived from AA are more pro-inflammatory in nature compared to those formed from EPA (3 series PGs and TXs and 5 series LTs are from EPA). Plasma and tissue levels of PGE2 are elevated in many inflammatory conditions {reviewed in [23,30,41,42,175,176,177]}. It is known that once the inflammatory process reaches its peak and so also the local concentrations of PGE2, the metabolism of AA is redirected to form LXA4 to trigger inflammation resolution process. This redirection of AA to form LXA4 to induce resolution of inflammatory process depends on the local actions of IL-1β, IL-4 and other anti-inflammatory cytokines (see Figure 3). This redirection of AA to form LXA4 is also dependent on the activation of COX-2 and LOX enzymes. When this switchover to form LXA4 from AA instead of PGE2 does not happen in a timely fashion it will lead to continuation of inflammation and chronicity of the underlying pathology as seen in obesity, type 2 diabetes mellitus, hypertension, lupus, RA and other chronic inflammatory conditions. PGE3 is less pro-inflammatory compared to PGE2, suggesting that it is unlikely that EPA metabolism behaves in the same fashion as that of AA (redirection of EPA metabolism to form resolvins instead of PGE3). This is supported by our observation that AA and LXA4 are more potent than EPA, resolvins, and protectins in preventing alloxan and streptozotocin-induced cytotoxicity to pancreatic β cells and diabetes mellitus [35,36]. Hence, it is proposed that AA is the principle fatty acid that is responsible for the switchover from proinflammatory state to anti-inflammatory state (by the formation of LXA4 from AA instead of PGE2), whereas the formation of resolvins, protectins and maresins from EPA and DHA serve more as supporters of the resolution process initiated by LXA4. AA supplementation does not enhance PGE2 but augments LXA4 formation when employed under inflammatory conditions [216,217,218]. AA can be injected intravenously without any side effects and is safe for human use [240]. It has been shown that oral supplementation of AA ~ 1000 mg/day to 2000 mg/day for 50 days is without any side effects [241,242,243,244]. Our studies showed that AA has potent anti-inflammatory actions and oral administration of AA leads to an increase in plasma LXA4 levels [35,36]. In view of these evidences, it is recommended that studies need to be performed using oral AA (and possibly GLA, DGLA, EPA and DHA) and various co-factors needed for optimal EFA metabolism in various diseases and whether such an intervention can postpone aging or lead to healthy aging. It remains to be seen to what extent this knowledge can be exploited to prevent or postpone aging and aging associated diseases.

## Figures and Tables

**Figure 1 biomolecules-11-00241-f001:**
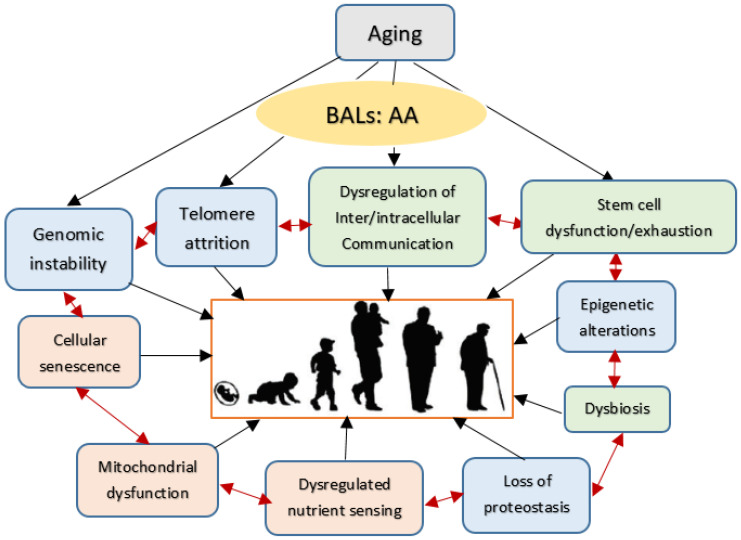
Scheme showing possible changes that can occur during ageing. Changes that are likely to be primary events in the causation of ageing are given in blue; responses to damage are given in orange and hallmarks of the effect of the alterations in genomic stability, telomere attrition and epigenetic changes and loss of proteostasis is given in green. Kindly note that all these events can interact with each other. In all the events associated with aging are modulated by AA (arachidonic acid, 20:4 n−6) and its metabolites.

**Figure 2 biomolecules-11-00241-f002:**
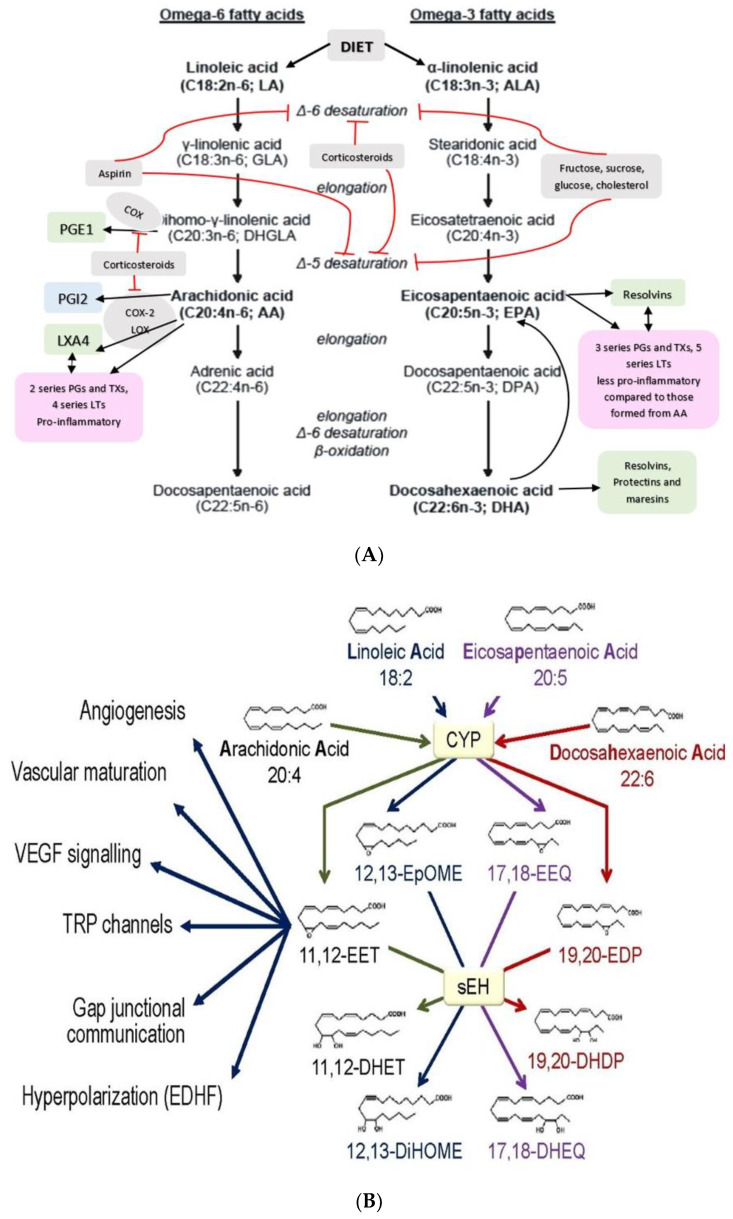
(**A**) Scheme showing metabolism of essential fatty acids and their important products. (**B**) Scheme showing metabolism of AA, eicosapentaenoic acid (EPA) and docosahexaenoic acid (DHA) by cytochrome P450 enzymes. DHA, alpha-linolenic acid (ALA), LXA4 and possibly, other bioactive lipids (BALs) may inhibit sEH enzyme and thus, bring about some of their beneficial actions. Some of the beneficial actions of metabolites of AA, EPA and DHA formed due to the action of cytochrome P450 enzymes (such as 11,12-EET) is also shown. (**C**) Metabolism of dihomo-gamma-linolenic acid (DGLA) by cytochrome P450 enzymes. (**D**) Scheme showing the formation of leukotrienes (LTs) and lipoxins (LXs) in various types of cells from arachidonic acid. Similar metabolism occurs regarding EPA and DHA to form resolvins, protectins and maresins. Note the conversion of 15S-H(p)ETE to LXA4 and LTA4 to LXA4/LXB4 by the action of 5-lipoxygenase (5-LO) and 12-lipoxygenase (12-LO). This could be one potential mechanism by which leukocytes and platelets at the site of inflammation can convert pro-inflammatory leukotrienes to anti-inflammatory lipoxins and thus, initiate resolution of inflammation.

**Figure 3 biomolecules-11-00241-f003:**
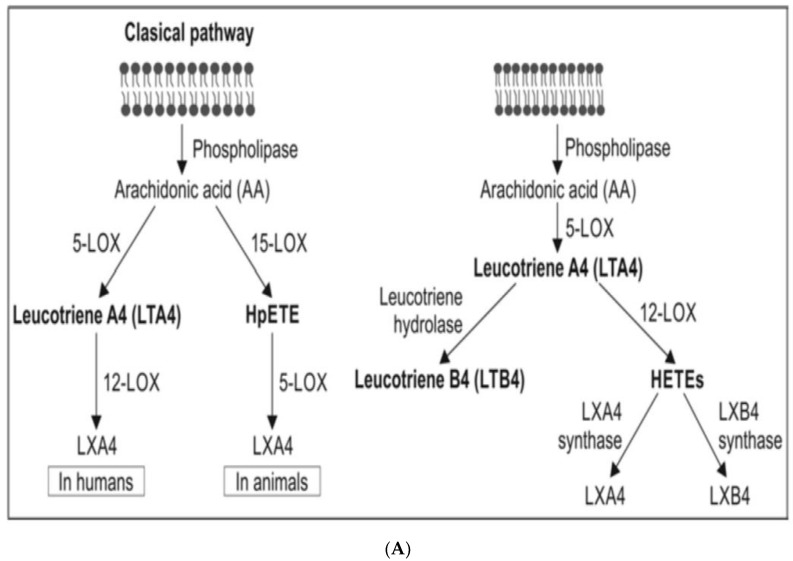
(**A**) Scheme showing the metabolism of AA and how pro-inflammatory LTA4 can be converted to anti-inflammatory LXA4. Similar conversion of pro-inflammatory PGs, TXs and LTs to anti-inflammatory resolvins, protectins and maresins may occur. (**B**) Scheme showing interaction(s) among BALs, cytokines, glucocorticoids, and inflammatory process. **(-)** Indicates inhibition of action or synthesis; **(+)** indicates increase in synthesis or action. sEH = soluble epoxide hydrolase; VNS = vagal nerve stimulation; ROS = reactive oxygen species. α-KG = alpha-ketoglutarate. Infections, surgery, and injury activate PLA2 leading to the release of AA and other unsaturated fatty acids from the cell membrane lipid pool. Simultaneously circulating leukocytes, macrophages and other immunocytes are activated that release IL-6 and TNF-α, which stimulate PLA2 leading to release of AA (and DGLA, EPA, DPA an DHA). COX-2 and LOX enzymes convert DGLA, AA, EPA and DHA to form their respective metabolites. To balance the action of pro-inflammatory cytokines, there will be release of anti-inflammatory cytokines IL-4 and IL-10. Pro-IL-β1 released is converted to IL-β1, a pro-inflammatory cytokine. IL-β1 acts on cPLA2/sPLA2 to induce the release of second wave of AA from the cell membrane that can be converted to LXA4, an anti-inflammatory molecule. During the first 24 h of injury, infection, and surgery, iPLA2 is activated. AA and other PUFAs released due to the activation of iPLA2 are utilized mainly to form pro-inflammatory bioactive lipids. In contrast to this, 48–96 h after infection, injury, and surgery activation of cPLA2/sPLA2 occurs that induce the release of second wave of AA and other PUFAs (polyunsaturated fatty acids) that is used to form anti-inflammatory LXA4, PGD2 and PGJ2. This switchover from pro-inflammatory PGE2 and LTs to anti-inflammatory LXA4 is facilitated by PGE2. Once the local tissue and plasma concentrations of PGE2 reach the peak, it (PGE2) activates 5-LOX and 15-LOX enzymes to enhance the formation of LXA4 to initiate resolution of inflammation and enhance wound healing. If PGE2 concentrations fail to reach its peak levels, stimulation of 5- and 15-LOX fails to occur, and so inflammation persists. It is likely that once the PGE2 levels reach their peak, it stimulates cPLA2 and sPLA2 enzymes to induce second wave of AA release. EPA, DHA and DGLA may have actions like AA (not shown in the figure). EPA forms the precursor to 3 series PGs, TXs and 5 series LTs (that are less pro-inflammatory compared to 2 series PGs, TXs and 4 series LTs formed from AA but are nevertheless pro-inflammatory) and resolvins of E series that are anti-inflammatory. DHA forms precursor to resolvins, protectins and maresins all of which are potent anti-inflammatory molecules. DGLA is the precursor of PGE1, an anti-inflammatory molecule, that has actions like LXA4 (see Table 1 and Figure 3C). This figure is modified from reference [37]. (**C**) Scheme showing the metabolism of DGLA and their metabolites. PGH1 has pro-inflammatory actions but is less potent compared to PGH2 and PGE2. PGE1 is anti-inflammatory in nature. (**D**) PGH1 stimulates Ca^2+^ mobilization in CRTH2 transfected and primary eosinophils. It is evident that PGH1 is less potent compared to PGH2 and so is less pro-inflammatory compared to PGH2 and PGE2. This data is taken from *PLoS ONE*
**2012**, *7*, e33329, doi:10.1371/journal.pone.0033329. (**E**) Summary of metabolites formed from AA, DPA, EPA and DHA and possible timeline of their formation during inflammation and resolution phases of inflammation. It may be noted that once the resolution process of inflammation starts, there is a need for removal of debris, regeneration of tissues (stem cells need to move to the site of inflammation, proliferate, differentiate and induce repair of damaged tissues and restore homeostasis). It is likely that lipoxins are needed for inhibition of inflammation; resolvins for resolution of inflammation; protectins for protecting normal tissues; and maresins for stem cell function. All these events may occur simultaneously but in a highly coordinated and regulated fashion. It is proposed that the concentrations of various molecules involved in inflammation and its resolution and restoration of homeostasis is as follows: 24 h:
PGE2↑↑↑↑; LXA4↑; RSVs↔; PRTs↔; MaRs↔. 48 h: PGE2↑↑↑; LXA4↑↑; RSVs↑; PRTs↑; MaRs↑. 72 h: PGE2↑↑; LXA4↑↑↑; RSVs↑↑; PRTs↑↑↑; MaRs↑↑↑. 96 h: PGE2↑; LXA4↑↑; RSVs↑↑↑; PRTs↑↑↑; MaRs↑↑↑↑. >96 h: PGE2↑; LXA4↑; RSVs↑↑; PRTs↑↑; MaRs↑↑↑. The actions of these compounds in the inflammation and wound healing process can be as follows: LXA4 → anti-inflammatory >resolution >protection >proliferation. RSVs → resolution > anti-inflammatory >protection >proliferation. PRTs → protection > resolution > anti-inflammatory > proliferation. MaRs → proliferation > protection > resolution > anti-inflammatory. Resolution refers to resolution of inflammation. Protection refers to protection of normal cells/tissues from injurious agents. Proliferation refers to proliferation of stem cells and other cells to replace damaged cells/tissues. Even though all compounds have similar and overlapping actions and possess anti-inflammatory properties, each lipid may show one particular action more compared to the other actions.

**Figure 4 biomolecules-11-00241-f004:**
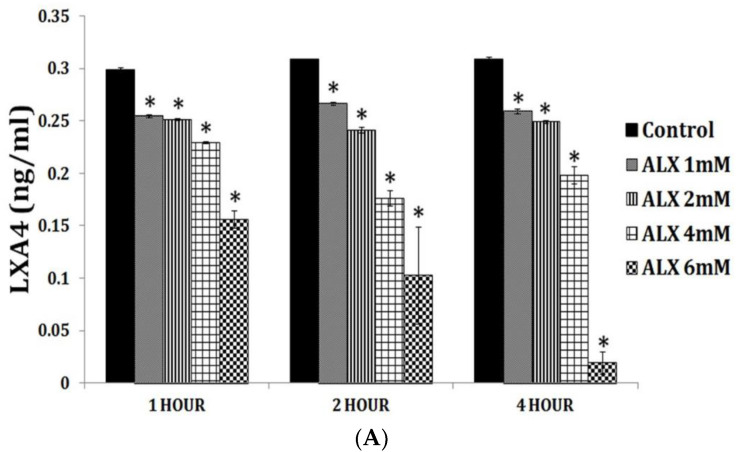
Effect of various polyunsaturated fatty acids (PUFAs) on LXA4 secretion by RIN cells (rat insulinoma cells) treated with alloxan and streptozotocin. Alloxan and streptozotocin-induced inhibition of LXA4 secretion by RIN cells is restored to near normal by GLA, AA, EPA and DHA compared to control. (**A**) RIN5F cells were treated with various doses (1, 2, 4, 6 mM) of alloxan for 1, 2, 4 h. The LXA4 was estimated by ELISA in the supernatant of cultures. (**B**) RIN5F cells were treated with 10 μg/mL GLA, AA, EPA and DHA and alloxan (6 mM) for 1 h. Streptozotocin (21 mM) treated RIN cells (for 24 h) were exposed to 10 μg/mL of various PUFAs (**C**). The LXA4 was estimated in the supernatant of the cell cultures. * *p* < 0.05 compared to untreated control, # *p* < 0.05 compared to alloxan, compared to STZ. It is seen that at 10 μg/mL dose of EPA and DHA treatment there is no increase in LXA4 secretion by RIN5F cells in vitro in the presence of alloxan (6 mM) (Figure 4A). However, when RIN5F cells were supplemented with 15 μg/mL of EPA and DHA there is a significant increase LXA4 secretion even in the presence of alloxan (Figure 4B). In contrast 10 μg/mL of PUFAs could increase LXA4 secretion to near normal by RIN cells (Figure 4C) (AA > GLA > EPA > DHA). It is seen from this data that GLA, EPA and DHA can augment LXA4 formation but are less potent compared to AA. This suggests that some of the anti-inflammatory actions of GLA, EPA and DHA could be due to their action to enhance LXA4 formation in addition to their ability to give rise to PGE1 (from GLA); resolvins of E series from EPA and resolvins of D series, protectins and maresins from DHA. This data is taken from references [35,36].

**Figure 5 biomolecules-11-00241-f005:**
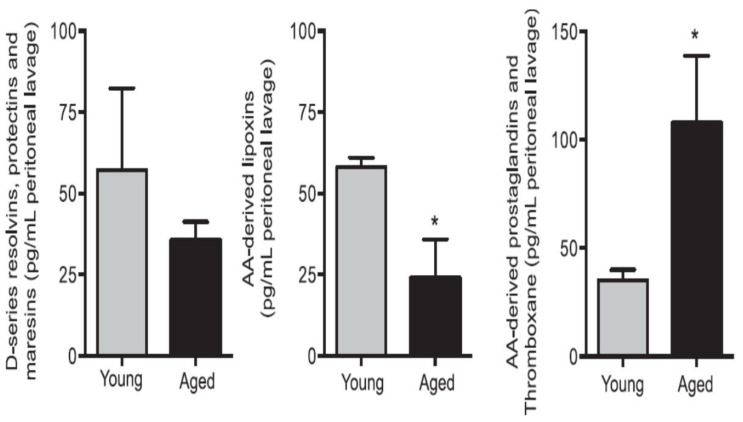
Aged mice show reduced resolvins, protectins and maresins and LXA4 in the peritoneal lavage of zymosan challenged animals. * *p* < 0.05 compared to young mice. This data is taken from reference no. 73.

**Figure 6 biomolecules-11-00241-f006:**
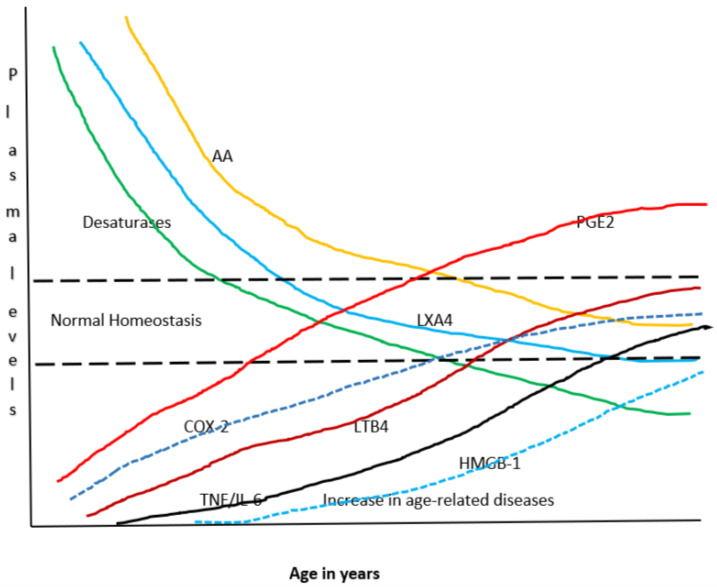
Scheme showing potential relationship among AA, PGE2, LXA4, desaturases and cytokines with age. It can be seen that with advancing age there is a gradual decrease in the activity of desaturases and a steady fall in the concentrations of AA and LXA4 and a gradual increase in that of PGE2, LTB4 and TNF-α and IL-6. Thus, with advancing age there is gradual and steady increase in pro-inflammatory status and an increase in age-related diseases and a decline in the ability of tissues/cells/organs/humans to fight or ameliorate inflammation due to a decline in anti-inflammatory molecules/capacity. It is envisaged that under normal physiological conditions a delicate balance is maintained among all these molecules/enzymes to maintain homeostasis.

**Figure 7 biomolecules-11-00241-f007:**
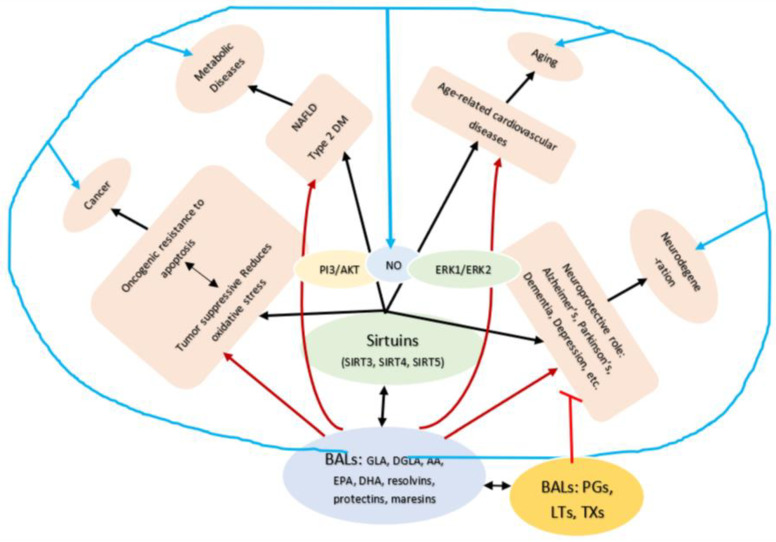
Scheme showing potential relationship between SITs (sirtuins) and BALs and their role in various age-related diseases. For details see text.

**Table 1 biomolecules-11-00241-t001:** Fatty acid composition phospholipid fraction of different types of cells/tissues of Wistar rats. This data is taken from Manjari, V., Das, U. N. (2000) Prostaglandins Leukot. Essen. Fatty Acids. 62, 85–96., and Mohan IK, Das UN. (2001) Nutrition. 17, 126–151 (references 1 and 2 under references section).

Fatty Acid	Plasma	Liver	Stomach	Duodenum	Skeletal Muscle
14:0 (MA)	0.39 ± 0.1	0.14 ± 0.04	0.49 ± 0.08	1.26 ± 0.37	-
16:0 (PA)	36.84 ± 1.85	27.6 ± 0.49	27.16 ± 4.59	27.59 ± 3.9	23.66 ± 2.19
18:0 (SA)	14.55 ± 3.52	11.67 ± 1.75	18.22 ± 2.93	8.17 ± 1.24	15.54 ± 1.35
18:1 n−9 (OA)	3.74 ± 0.53	2.48 ± 0.37	14.09 ± 2.51	7.66 ± 1.86	9.79 ± 1.43
18:2 n−6 (LA)	15.44 ± 1.44	11.27 ± 1.76	14.51 ± 3.41	8.54 ± 1.22	19.66 ± 1.79
18:3 n−6 (GLA)	0.22 ± 0.07	0.24 ± 0.02	-	-	0.42 ± 0.06
20:3 n−6 (DGLA)	-	0.24 ± 0.02	-	-	0.42 ± 0.05
20:4 n−6 (AA)	22.83 ± 2.3	32.6 ± 1.70	14.12 ± 1.82	16.09 ± 1.04	20.09 ± 2.42
18:3 n−3 (ALA)	0.22 ± 0.07	0.33 ± 0.09	0.42 ± 0.25	0.51 ± 0.07	0.50 ± 0.08
20:5 n−3 (EPA)	0.33 ± 0.08	0.59 ± 0.08	0.32 ± 0.02	0.35 ± 0.02	0.49 ± 0.05
22:6 n−3 (DHA)	1.94 ± 0.24	1.76 ± 0.41	1.0 ± 0.08	1.08 ± 0.2	4.14 ± 0.49

**Table 2 biomolecules-11-00241-t002:** Comparison between PGE1 and LXA4.

Property/Action	LXA4	PGE1
Derived from	Arachidonic acid (AA)	Di-homo-gamma-linolenic acid (DGLA)
Rate limiting step in AA/DGLA synthesis	Delta-6- and delta-5-desaturases	Delta-6-desaturase
Platelet anti-aggregator	++	+
Vasodilator	++	+
Anti-inflammatory action	++	+
Suppresses IL-6 and TNF-α	++	+
Cytoprotective action	+++	++
Geno-protective action	+/−	++
Anti-diabetic action	++	+
Suppresses ROS generation	++	+
Suppression of PGE2 production	++	Not known
Inflammation resolution action	++	+
Wound healing action	++	++
Blood pressure lowering action	+	+
Anti-arrhythmic action	+/−	+/−
Protects endothelium	++	++
PGE2 can trigger synthesis	Yes	Not known
Anti-microbial action	++	+
Has a specific receptor	Yes-ALX	Yes-EP1 and EP3
Half-life	Few seconds	5–30 min

**Table 3 biomolecules-11-00241-t003:** Fatty acid analysis of the plasma PL (phospholipid) fraction in patients with pneumonia, septicemia, RA and lupus. Both pneumonia and sepsis are more common elderly. All values are expressed as mean ± S. E. This data is taken form [106]. ** *p* < 0.001 compared to control; * *p* < 0.05 compared to control.

Fatty Acid	Control (n = 10)	Pneumonia (n = 12)	Septicemia (n = 14)	RA (n = 12)	SLE (Lupus) (n = 5)
16:0	24.8 ± 3.4	32.5 ± 3.6	26.95 ± 4.1	30.2 ± 3.0	32.0 ± 3.75
18:0	23.3 ± 4.1	21.4 ± 7.1	24.58 ± 6.0	19.0 ±6.1	14.6 ± 5.82
18:1 n−9	13.1 ± 2.3	15.6 ± 3.2	16.5 ± 3.3 *	14.8 ± 2.1	16.0 ± 2.78
18:2 n−6	17.7 ± 3.1	14.2 ± 0.3 *	16.3 ± 2.4	17.5 ± 2.7	20.8 ± 2.2
18:3 n−6	0.13 ± 0.09	0.13 ± 0.08	0.04 ± 0.05 *	0.02 ± 0.04 **	0.01 ± 0.01 **
20:3 n−6	3.2 ± 0.79	1.5 ± 0.4 *	0.46 ± 0.54 *	2.5 ± 0.58	2.12 ± 0.52
20:4 n−6	8.8 ± 2.0	5.1 ±0.4 *	5.8 ± 1.6 *	9.5 ± 2.2	8.93 ± 2.0
22:4 n−6	0.42 ± 0.23	0.8 ± 0.9	0.34 ± 0.28	0.26 ± 0.37 **	0.18 ± 0.18 **
22:5 n−6	0.73 ± 0.55	0.45 ± 0.63	1.5 ± 1.02 *	0.6 ± 0.7	0.8 ± 1.0
18:3 n−3	0.27 ± 0.12	0.09 ± 0.04 *	0.16 ± 0.11 *	0.12 ± 0.16 *	0.1 ± 0.1 *
20:5 n−3	0.25 ± 0.26	0.23 ± 0.24	0.01 ± 0.01 *	0.05 ± 0.14 **	0.04 ± 0.04 **
22:6 n−3	1.43 ± 0.43	0.54 ± 0.43 *	1.2 ± 1.14	0.62 ± 0.56 *	0.88 ± 0.75 *

**Table 4 biomolecules-11-00241-t004:** Content of fatty acids in normal liver, hepatoma cells, and in microsomal suspensions from normal liver and Yoshida hepatoma cells. All values of mean ± S. E.

Measurement (Fatty Acid)	Normal Intact Liver	Intact Yoshida Cells	Normal Liver Microsomes	Yoshida Microsomes
16:0	18.5 ± 0.2	18.7 ± 2.0	18.9 ± 1.1	18.5 ± 0.5
18:0	17.5 ± 0.5	13.3 ± 1.1	22.0 ± 3.0	13.7 ± 0.2
18:1, n−9 (oleic acid)	12.1 ± 1.0	21.5 ± 0.8	8.6 ± 1.0	18.1 ± 0.3
20:4 (AA)	16.7 ± 2.4	8.7 ± 0.7	19.1 ± 2.4	9.6 ± 0.8
22:5	-	2.9 ± 0.1	-	2.4 ± 0.3
22:6 (DHA)	6.3 ± 0.2	5.2 ± 0.6	6.1 ± 0.3	5.3 ± 0.4

**Table 5 biomolecules-11-00241-t005:** The percentage of distribution of fatty acids from plasma phospholipid fraction in patients with hypertension (HTN), coronary heart disease (CHD), type 2 diabetes mellitus, and diabetic nephropathy that are common with advanced age.

Fatty Acid	Control	HTN	CHD	Type 2 DM	Diabetic Nephropathy
16:0	25.9 ± 3.0	29.3 ± 2.7 *	27.8 ± 3.5	26.6 ± 5.2	26.8 ± 2.7
18:0	20.9 ± 3.6	23.2 ± 4.9 *	18:0 ± 10.7	14.6 ± 4.1	11.6 ± 3.6 *
18:1 n−9	13.0 ± 2.3	12.1 ± 1.5	11.5 ± 3.1	12.0 ± 2.6	14.5 ± 3.1
18:2 n−6 (LA)	18.6 ± 3.1	14.5 ± 3.1 *	17.8 ± 5.0	13.9 ± 5.3	15.1 ± 3.1
18:3 n−6 (GLA)	0.14 ± 0.1	0.4 ± 0.3 *	0.1 ± 0.1 *	0.2 ± 0.3	0.1 ± 0.2
20:3 n−6 (DGLA)	3.4 ± 1.0	3.1 ± 0.9	2.7 ± 1.1	1.7 ± 1.0 *	2.0 ± 0.8 *
20:4 n−6 (AA)	9.4 ± 1.8	7.8 ± 2.0 *	7.0 ± 2.1 *	4.6 ± 1.8 *	6.6 ± 2.6 *
22:5 n−6	0.7 ± 0.4	0.4 ± 0.4 *	1.0 ± 0.9	2.1 ± 0.6 *	1.3 ± 0.5 *
18:3 n−6/18:2 n−6	0.008	0.026	0.005	0.017	0.008
20:4 n−6/18:2 n−6	0.51	0.54	0.39	0.33	0.43
20:4 n−6/20:3−6	2.8	2.53	2.59	2.8	3.3
18:3 n−3 (ALA)	0.2 ± 0.1	0.4 ± 0.2 *	0.3 ± 0.5	0.1 ± 0.2 *	0.1 ± 0.1 *
20:5 n−3 (EPA)	0.4 ± 0.4	0.6 ± 0.6	0.1 ± 0.2 *	0.3 ± 0.3	0.2 ± 0.3
22:5 n−3	0.5 ± 0.2	0.4 ± 0.5	0.3 ± 0.3 *	1.6 ± 1.3	1.7 ± 1.1
22:6 n−3 (DHA)	1.4 ± 0.5	1.2 ± 0.6	0.8 ± 0.4 *	0.5 ± 0.4 *	0.5 ± 0.3 *
20:5 n−3/18:3 n−3	1.8	1.39	0.41	3.2	4.0

All values are expressed as mean ± S. D. * *p* < 0.05 compared to control. This data is taken from [155].

## Data Availability

All the data is provided in the manuscript.

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
