# Peer review of "“Cell Membrane Theory of Senescence” and the Role of Bioactive Lipids in Aging, and Aging Associated Diseases and Their Therapeutic Implications"

_biomolecules, 2021, doi:10.3390/biom11020241_

Round 1

Reviewer 1 Report

The review of Undurti N Das deals with the changes in lipid composition during aging and the impact of lipid in diseases. Lipids are increasingly in the focus of various research in recent years and are strongly associated with many diseases. Unquestionably this article is interesting for a broad readership, especially for scientists dealing with lipids, but also e.g. neurodegeneration, geriatrics and nutrition. In my opinion it is in the clear focus of the journal. It has a clear focus, it is mostly comprehensive (see remarks in detail) and has a nice structure. It has to be pointed out that in some points the author trends to be more an opinion article and speculates, however this sections are mostly clearly visible as an opinion and are therefore from my side suitable in the context of this article. I have some comments before this article is suitable for publication.

- Many aspects are simplified too much resulting in some inaccuracies. Just as an example: “… DHA (docosahexaenoic acid) are derived from dietary linoleic acid (LA) and alpha-linolenic acid (ALA) by the action of desaturases whose activity declines with age.“ It is correct that desaturases are able to convert other PUFAs in DHA. However, main parts of DHA are taken up by food exogenously and not produced in the human body. A decline in DHA is mostly due to a malnutrition and not to a reduced desaturase activity. It is clear what the author wants to point out, however putting it like this it is not really correct. Please go carefully through the manuscript and check for this inaccuracies.

- The author mainly focuses on fatty acids ignoring important changes e.g. of gangliosides, sphingolipids, cholesterol, plasmalogens, during aging and especially during disease. At least a small section should be added referring to these important lipids as well.

-  The author nicely summarizes general mechanisms e.g. aspects of PGC1alpha or PLA2 activity. Unfortunately, although these important mechanisms are mentioned and explained, the context in respect to neurodegenerative diseases are not mentioned or explained. E.g. Sanchez Maja has nicely published the role of PLA2 in Alzheimer’s disease (AD). Or PGC1alpha and its regulation in AD is well known. I recommend to add a section dealing with AD, Parkinson disease or some other disease taking up these mechanisms and discussing them in the context of the diseases more clearly.

- In line with the first point some figures are not completely correct as well. E.g. Figure 7 states that AA is neuroprotective. This can be discussed very controversially. As it is known to increase inflammation, increase SMase activity etc.

Minor points

- The manuscript could benefit from having a native speaker go through it.

- some graphics seem to be “hand painted“ probably a vector based graphic program like corel draw or something comparable would help …

Summing it up it is a nice review. A little more effort should be invested to emphasize the link to some diseases and to avoid mistakes or inaccuracies, However I think this can be done in a reasonable time.

Author Response

The review of Undurti N Das deals with the changes in lipid composition during aging and the impact of lipid in diseases. Lipids are increasingly in the focus of various research in recent years and are strongly associated with many diseases. Unquestionably this article is interesting for a broad readership, especially for scientists dealing with lipids, but also e.g. neurodegeneration, geriatrics and nutrition. In my opinion it is in the clear focus of the journal. It has a clear focus, it is mostly comprehensive (see remarks in detail) and has a nice structure. It has to be pointed out that in some points the author trends to be more an opinion article and speculates, however this sections are mostly clearly visible as an opinion and are therefore from my side suitable in the context of this article. I have some comments before this article is suitable for publication. - Many aspects are simplified too much resulting in some inaccuracies. Just as an example: “… DHA (docosahexaenoic acid) are derived from dietary linoleic acid (LA) and alphalinolenic acid (ALA) by the action of desaturases whose activity declines with age.“ It is correct that desaturases are able to convert other PUFAs in DHA. However, main parts of DHA are taken up by food exogenously and not produced in the human body. A decline in DHA is mostly due to a malnutrition and not to a reduced desaturase activity. It is clear what the author wants to point out, however putting it like this it is not really correct. Please go carefully through the manuscript and check for this inaccuracies.

Response: In response to the reviewer’s suggestion, this section has been modified on page: 1 (and highlighted in red).

 - The author mainly focuses on fatty acids ignoring important changes e.g. of gangliosides, sphingolipids, cholesterol, plasmalogens, during aging and especially during disease. At least a small section should be added referring to these important lipids as well.

Response: As suggested by the reviewer, a separate para (page 280 has been outlined about the role of these lipids in aging.  

- The author nicely summarizes general mechanisms e.g. aspects of PGC1alpha or PLA2 activity. Unfortunately, although these important mechanisms are mentioned and explained, the context in respect to neurodegenerative diseases are not mentioned or explained. E.g. Sanchez Maja has nicely published the role of PLA2 in Alzheimer’s disease (AD). Or PGC1alpha and its regulation in AD is well known. I recommend to add a section dealing with AD, Parkinson disease or some other disease taking up these mechanisms and discussing them in the context of the diseases more clearly.

Response: As rightly suggested by the reviewer, a separate section on AD and BALs has been added on page 24-27.

- In line with the first point some figures are not completely correct as well. E.g. Figure 7 states that AA is neuroprotective. This can be discussed very controversially. As it is known to increase inflammation, increase SMase activity etc.

Response: As suggested by the reviewer, to make the arguments made by the author in the text clear, Figure 7 has been modified.

Minor points - The manuscript could benefit from having a native speaker go through it. 2 - some graphics seem to be “hand painted“ probably a vector based graphic program like corel draw or something comparable would help … Summing it up it is a nice review. A little more effort should be invested to emphasize the link to some diseases and to avoid mistakes or inaccuracies, However I think this can be done in a reasonable time.

Response: As suggested by the reviewer, the manuscript has been thoroughly read and revised and any minor errors have been corrected.

Reviewer 2 Report

Review

The subject of this review is the role the composition of the cell membrane plays in senescence and aging. This is an interesting topic, and the author has published extensively in many areas of health related to essential fatty acids. The paper is well-written, and the information supported by references. That said, the paper is very, very long and unfocused, and spends a lot of time on the basics of BAL, and not on their specific role in aging.

Figure 1 summarizes the myriad ways bioactive lipids may participate in aging. This is not surprising as there are BAL is every cell in the body, and they participate extensively in cellular functions. What would be helpful is a section on the extent to which the EFA content of specific cell types can be modified by diet, as the author does point out that linoleic acid and linolenic acid are essential nutrients, and tissue composition can be affected the presence of these EFA in the diet as well as their elongation and desaturation products.

On line 79 the section is titled ‘Cell membrane theory of aging’ which last until line 158. There are 11 points made about the composition of cell membranes which are presumably added to set up the ‘theory’ yet it is not presented. This is followed by the sentences….

‘The cell membrane is crucial for all the cellular functions. But it is not clear how one can correlate the cell membrane integrity and composition to changes seen with aging.

According to the title, correlating the cell membrane composition with aging is what the reader expects and now we are told it is not clear?  

The main point supporting the author’s view is that the proportion of BALs in cell membrane decrease as a function of aging, and the suggestion is put forward that increasing their levels may counteract the aging process. Little evidence is put forward to support this hypothesis. They must be studies that have compared erythrocyte BAL composition as a function of diet and age. This data would possibly support the author’s case.

Several hypotheses exist that tie EFA and BAL lipid composition to the aging process, and it is surprising the author does not address these. EFA and the BAL composition increases the susceptibility of membranes to lipid peroxidation, and AJ Hulbert has published several papers showing the BAL content of membranes is inversely proportional to lifespan across species, and that exceptionally long-lived species have lower EFA contents in their membranes (i.e. naked mole rats). The dissonance between this hypothesis and that of the author should be addressed. Additionally, the Membrane Hypothesis of Aging (MHA), which is related to the Free Radical Theory of Aging, proposes that lipid peroxidation and protein cross linking are causative factors in aging, and both would be expected to be increased in membranes with a higher content of PUFA.

This manuscript would be improved with cellular, animal, or human ecologic data that supports the hypothesis mechanistically, rather than extensive discussions of the BAL pathways. In addition, the two alternative hypotheses that link oxidative susceptibility of PUFA to aging need to be addressed.

Author Response

The subject of this review is the role the composition of the cell membrane plays in senescence and aging. This is an interesting topic, and the author has published extensively in many areas of health related to essential fatty acids. The paper is well-written, and the information supported by references. That said, the paper is very, very long and unfocused, and spends a lot of time on the basics of BAL, and not on their specific role in aging.

Response: I wish to thank the reviewer for his/her frank opinion and criticism. It is my opinion that it is difficult for any reader to understand a new topic such as bioactive lipids and their potential role in many human diseases. Hence, considerable effort has been spent on describing the basics of bioactive lipids and their metabolism. Having said this, it is also difficult for many to appreciate anew concept that as overwhelming as aging process in terms of BALs. This is clear form the fact that in general, it is believed by many that aging is due to free radical damage of tissues and DNA. But the evidence obtained from naked mole rat are completely contrary to this-they have low concentrations of antioxidants, high levels of lipid peroxides and have low incidence of cancer. This makes difficult reading and understanding. This aspect has been discussed in detail on page 30.   

Figure 1 summarizes the myriad ways bioactive lipids may participate in aging. This is not surprising as there are BAL is every cell in the body, and they participate extensively in cellular functions. What would be helpful is a section on the extent to which the EFA content of specific cell types can be modified by diet, as the author does point out that linoleic acid and linolenic acid are essential nutrients, and tissue composition can be affected the presence of these EFA in the diet as well as their elongation and desaturation products.

Response: As suggested by the reviewer, table 1 is provided in which the fatty acid composition of different tissues is given. This data shows that different tissues have different fatty acid composition implying that different tissues from the same individual or animal have different fatty acid composition.  

On line 79 the section is titled ‘Cell membrane theory of aging’ which last until line 158. There are 11 points made about the composition of cell membranes which are presumably added to set up the ‘theory’ yet it is not presented. This is followed by the sentences…. ‘The cell membrane is crucial for all the cellular functions. But it is not clear how one can correlate the cell membrane integrity and composition to changes seen with aging. According to the title, correlating the cell membrane composition with aging is what the reader expects and now we are told it is not clear? The main point supporting the author’s view is that the proportion of BALs in cell membrane decrease as a function of aging, and the suggestion is put forward that increasing their levels may counteract the aging process. Little evidence is put forward to support this hypothesis. They must be studies that have compared erythrocyte BAL composition as a function of diet and age. This data would possibly support the author’s case. Several hypotheses exist that tie EFA and BAL lipid composition to the aging process, and it is surprising the author does not address these. EFA and the BAL composition increases the susceptibility of membranes to lipid peroxidation, and AJ Hulbert has published several papers showing the BAL content of membranes is inversely proportional to lifespan across 3 species, and that exceptionally long-lived species have lower EFA contents in their membranes (i.e. naked mole rats). The dissonance between this hypothesis and that of the author should be addressed. Additionally, the Membrane Hypothesis of Aging (MHA), which is related to the Free Radical Theory of Aging, proposes that lipid peroxidation and protein cross linking are causative factors in aging, and both would be expected to be increased in membranes with a higher content of PUFA. This manuscript would be improved with cellular, animal, or human ecologic data that supports the hypothesis mechanistically, rather than extensive discussions of the BAL pathways. In addition, the two alternative hypotheses that link oxidative susceptibility of PUFA to aging need to be addressed.

Response: As suggested by the reviewer, tables 1,3 and 4 and Figures 5 and 6 are provided that show how the fatty acid composition changes with tissue, disease and data pertaining to fatty acid changes with age has been discussed in the text of the manuscript (see pages 23-24.)

Round 2

Reviewer 1 Report

The authors have addressed all my concerns and remarks. In my opinion the manuscript is now suitable for publication in its present form

Author Response

Thank you very mush for your positive reply.